# Quantitative impacts of vertical transport on long-term trend of nocturnal ozone increase over the Pearl River Delta region during 2006-2019

Yongkang Wu[1,2], Weihua Chen[1,2,*], Yingchang You[1,2], Qianqian Xie[1,2], Shiguo Jia[3], Xuemei Wang[1,2]

[1]Institute for Environmental and Climate Research, Jinan University, Guangzhou, 510632, P. R. China

[2]Guangdong-Hongkong-Macau Joint Laboratory of Collaborative Innovation for Environmental Quality, Guangzhou 511443, China

[3]School of Atmospheric Sciences, Sun Yat-sen University and Southern Marine Science and Engineering Guangdong Laboratory (Zhuhai), Zhuhai, 519082, China

*Correspondence:* Weihua Chen (chenwh26@163.com)

**Abstract.** The Pearl River Delta (PRD) region in southern China has been subject to severe ozone ($O_3$) pollution during daytime, and anomalous nocturnal $O_3$ increase (NOI) during nighttime. In this study, the spatiotemporal variation of NOI events in the PRD region from 2006 to 2019 is comprehensively analyzed and the role of vertical transport in the occurrence of NOI events is quantified based on observed surface and vertical $O_3$ and the fifth-generation European Centre for Medium-Range Weather Forecasts (ECMWF) reanalysis (ERA5) dataset. The results show that the average annual frequency of NOI events in the whole PRD region during the 14-year period is estimated to be $53 \pm 16$ d yr$^{-1}$, with an average of $58 \pm 11$ µg m$^{-3}$ for the nocturnal $O_3$ peak (NOP) concentration. Low-level jets (LLJs) are the main meteorological processes triggering NOI events, explaining on average 61% of NOI events. Annual NOI events exhibit an upward trend before 2011 (4.70 d yr$^{-1}$) and a downward trend thereafter (-0.72 d yr$^{-1}$), which is consistent with the annual variation of LLJs (r=0.88, p<0.01). Although the contribution of convective storms (Conv) to NOI events is relatively small with an average value of 11%, Conv-induced NOI events steadily increased at a rate of 0.26 d yr$^{-1}$ during this 14-year period due to the impact of urbanization. Seasonally, relatively higher frequency of NOI events is observed in spring and autumn, which is consistent with the seasonal pattern of LLJs and maximum daily 8-h average (MDA8) $O_3$. Spatially, NOI events are frequent in the eastern PRD, which agrees well with the spatial distribution of the frequency of LLJs and partially overlaps with the distribution of MDA8 $O_3$ concentration, suggesting that vertical transport plays a more important role in NOI events than daytime $O_3$ concentration. The WRF-CMAQ model and the observed vertical $O_3$ profiles are further applied to illustrate the mechanisms

of NOI formation caused by LLJs and Conv. The results confirm that both LLJs and Conv trigger NOI
events by inducing downdrafts with the difference being that LLJs induce downdrafts by wind shear
while Conv by compensating downdrafts. Through observational and modeling analysis, this study
presents the long-term (2006-2019) trends of NOI events in the PRD region and quantifies the
contribution of meteorological processes for the first time, emphasizing the importance of vertical
transport as well as daytime $O_3$ concentration for the occurrence of NOI events.
**Keywords:** Nocturnal ozone increase; Ozone profile; Low-level jets; Convective storms; Pearl River
Delta; Long-term trend
**1 Introduction**
As a secondary pollutant, surface ozone($O_3$) is formed via photochemical reactions involving nitric oxide
($NO_x$) and volatile organic compounds (VOCs) in the presence of sunlight. Therefore, $O_3$ shows
significant diurnal variation, with concentration peaks observed during daytime (Kleinman et al., 1994;
Zhang et al., 2004). During nighttime, $O_3$ production ceases owing to the absence of sunlight, and dry
deposition and NO titration (Eq. (1)) remove $O_3$ directly from the atmosphere, lead to relatively low $O_3$
concentrations at night (Jacob, 2000; Brown et al., 2006).
$$NO + O_3 \rightarrow NO_2 + O_2 \tag{1}$$
However, $O_3$ concentrations do not always remain at low levels during nighttime, and frequent nocturnal
$O_3$ increase (NOI) events have been observed in various countries in Asia, Europe, North America, etc.
in different topographies (plains, valleys, mountains, etc.) (Kuang et al., 2011; Kulkarni et al., 2013;
Klein et al., 2019; Udina et al., 2019; Zhu et al., 2020). Kulkarni et al. (2015) found that NOI events were
observed around 03:00 (LT) in the UK, with concentrations as high as 118 μg m$^{-3}$, which was much
higher than the monthly average daytime $O_3$ concentration (69 ± 10 μg m$^{-3}$). Yusoff et al. (2019) also
reported that frequent NOI events were observed in some cities in Malaysia, and the annual trend of
nocturnal $O_3$ concentration was found to on the increase based on 11 years of ground-based
measurements. High nocturnal surface $O_3$ concentrations have adverse effects on crops and vegetation,
leading to plant water loss, stomatal sluggishness, and reduction in plant production (Caird et al., 2007;
Cirelli et al., 2016; Yue et al., 2017) as well as on human health (Kurt et al., 2016; Carré et al., 2017).
Because there is no photochemical production of $O_3$ at night, NOI events are likely to be due to
meteorological processes (Salmond and McKendry, 2002). It has been widely recognized that low-level
jets (LLJs) are one of the most important meteorological processes that cause NOI events (Salmond and
McKendry, 2002; Kuang et al., 2011; Sullivan et al., 2017). After sunset, radiative cooling and
subsequent weakened turbulence result in a stratified nocturnal boundary layer (NBL) with an altitude of
400-500 m (Stull, 1988; Sugimoto et al., 2009; Fan et al., 2022). A residual layer (RL) exists above the
NBL, which contains residual $O_3$ produced during daytime. When an LLJ occurs during nighttime, it can
break the delamination between the NBL and the RL by wind shear and bring the $O_3$ from the RL to the
surface, leading to an accumulation of ground-level $O_3$. An analysis of aircraft data from California has
shown that LLJs promote the mixing between the NBL and the RL and transport $O_3$ from the RL to the
surface, leading to NOI events (Caputi et al., 2019). Convective storms (Conv) are another
meteorological process that contributes to NOI events, especially at the equator and in tropical areas that
have a higher frequency of convection (Prtenjak et al., 2013; Zhu et al., 2020; Wu et al., 2020). Dias-
Junior et al. (2017) revealed that downdrafts induced by Conv play an important role in triggering NOI
events in the Amazon region of Brazil based on 1-yr observations. Jain et al. (2007) noted that NOI
events in India are often accompanied by thunderstorms and stable boundary layer conditions. Other
meteorological processes that are highly dependent on topography, such as sea-land breezes and
mountain-valley breezes, also contribute to NOI events (Salmond and McKendry, 2002; Nair et al., 2002).
Seibert et al. (2000) pointed out that nocturnal $O_3$ concentrations are elevated during foehn events in the
Eastern Alps.
Therefore, NOI events are not an exception and can occur worldwide as a result of certain meteorological
processes (LLJs, thunderstorms, foehn, etc.) (Hu et al., 2013; Caputi et al., 2019; Klein et al., 2019; Udina
et al., 2019; Shith et al., 2021). LLJs and Conv are important factors influencing the generation of NOI
events; however, their relative contribution to NOI events has not yet been quantified. Most previous
studies are focused on the analysis of a single NOI event or NOI events at limited monitoring sites for
short periods (Jain et al., 2007; Hu et al., 2013; He et al., 2021). Consequently, it is of great importance
to investigate the long-term trends of NOI events on a larger scale to further quantify the impacts of
meteorological processes, such as LLJs and Conv, on NOI events.
In China, ground-level $O_3$ pollution has been deteriorated in recent years, especially in the Pearl River
Delta (PRD) region (Wang et al., 2017). Liao et al. (2021) investigated ozonesonde profiles recorded in
Hong Kong during 2000-2019 and indicated that $O_3$ concentrations in the lower troposphere have
increased substantially at a rate of 0.618 ppbv $yr^{-1}$, indicating a continuous deterioration of $O_3$ pollution
in the PRD region over the last 20 years. The PRD region is the first urban agglomeration in China to
change its main pollutant from particulate matter with an aerodynamic diameter of less than or equal to
2.5 mm ($PM_{2.5}$) to $O_3$. Numerous studies in the PRD region have investigated the daytime $O_3$
characteristics, such as the long-term trends (Xue et al., 2014; Li et al., 2022), the nonlinear response of
$O_3$ to precursor emissions (Lu et al., 2010; Mao et al., 2022), the source apportionment of $O_3$ (Shen et
al., 2015; Liu et al., 2020), and the relative contributions of precursor emissions and meteorology to $O_3$
(Yang et al., 2019b; Chen et al., 2020). In terms of nighttime $O_3$, Tong and Leung (2012) observed a
double-peak pattern of diurnal $O_3$ variation in Hong Kong during 1990-2005, and found that nocturnal
$O_3$ peaks are sometimes higher than daytime maxima. He et al. (2021) studied an NOI event at the city
Shaoguan in Guangdong Provine, and found that nocturnal mountain-valley breezes from the Nanling
Mountains transported $O_3$ from the RL to the surface. However, studies on the spatio-temporal
distribution of nocturnal $O_3$ concentration in the PRD region and the factors influencing it are still lacking.
There is an urgent need to comprehensively study the characteristics of NOI events in the PRD region as
it is frequently affected by special meteorological processes (such as LLJs and Conv) that favor NOI
events due to its special topography with the coast to the south and the mountains to the north. In addition,
high population densities and increasing number of people active at night in the PRD region make NOI
events an important potential risk to human health (Kurt et al., 2016; Carré et al., 2017; Yang et al.,
2019a; Zhang et al., 2021).
In this study, the long-term trends and spatial distribution of NOI are presented via in-situ hourly $O_3$
concentration data collected from 16 air quality monitoring sites in the PRD region during 2006-2019.
In addition, the relative contributions of LLJs and Conv to NOI events are quantified based on the ERA5
reanalysis dataset. Finally, the observed vertical profile of $O_3$ and the Weather Research and Forecasting
(WRF) model coupled with the Community Multiscale Air Quality (CMAQ) model are applied to further
elaborate the impacts of LLJs and Conv on the selected typical NOI events. This study provides a
comprehensive analysis of NOI events and the meteorological factors influencing them in the PRD region
over a 14-year period for the first time, expanding our knowledge of the meteorological role in NOI
events.
**2 Data and methods**
**2.1 Data sources**
The dataset used in this study is summarised in Table 1. In brief, the observed hourly $O_3$ concentrations
at the 16 air quality monitoring sites in the PRD region from 2006 to 2019 are provided by the
Guangdong-Hong Kong-Macao Pearl River Delta Regional Air Quality Monitoring Network (HKEPD,
2017) (Fig. 1). More detailed information of these sites can be found in Table S1. The observed hourly
$O_3$ data were used for subsequent NOI and NOP analyses, and evaluation of $O_3$ simulations.
The vertical distribution of $O_3$ concentrations observed at the Dongguan superstation (23.02° N,113.79°
E) in 2019 is also used to investigate the impact of Conv on a particular NOI event. The vertical profile
of $O_3$ is measured using an $O_3$ lidar (Model: LIDAR-G-2000). The detection height of the $O_3$ lidar is 3
km, with a vertical spatial resolution of 7.5 m and a temporal resolution of 12 mins.
The observed meteorological variables at the 9 meteorological sites (Fig. 1) in the PRD region are
obtained from the Chinese National Meteorological Centre (CNMC, http://www.cma.gov.cn/, last
accessed on February 10, 2022), including temperature (T2), relative humidity (RH), and wind speed
(WS10). The observed meteorological data were used to evaluate the performance of the model. More
detailed information of the 9 meteorological sites can be found in Table S2.
To investigate the impacts of meteorological processes on NOI events, the ERA5 reanalysis dataset
(https://cds.climate.copernicus.eu/cdsapp#!/home, last accessed on February 10, 2022) provided by the
European Centre for Medium-Range Weather Forecasts (ECMWF) is used in this study. The ERA5
reanalysis dataset, which currently covers the period from 1979 to present, is provided on regular
latitude–longitude grids at approximately 0.25° × 0.25° and up to 1 h frequency. Vertically, ERA5
resolves the atmosphere using 137 levels from the surface to an altitude of 0.01 hPa. The performance of
ERA5 is evaluated in previous studies and has been shown to be adequate for further analysis (Olauson,
2018; Hersbach et al., 2020). The ERA5 reanalysis dataset includes wind speed, precipitation,
temperature, and vertical wind velocity. Since the ERA5 reanalysis dataset was gridded, the nearest-
neighbour interpolation method is used to obtain site-specific meteorological variables at the 16 air
quality monitoring sites.
The observed cloud-top temperature (CTT) data for 2019 obtained from the Fengyun-2G satellite
(http://satellite.nsmc.org.cn/, last accessed on August 31, 2022) are used to indicate the occurrence of
convection. The CTT data cover the East Asia region with a spatial resolution of 0.1° and the temporal
resolution of 1 h.

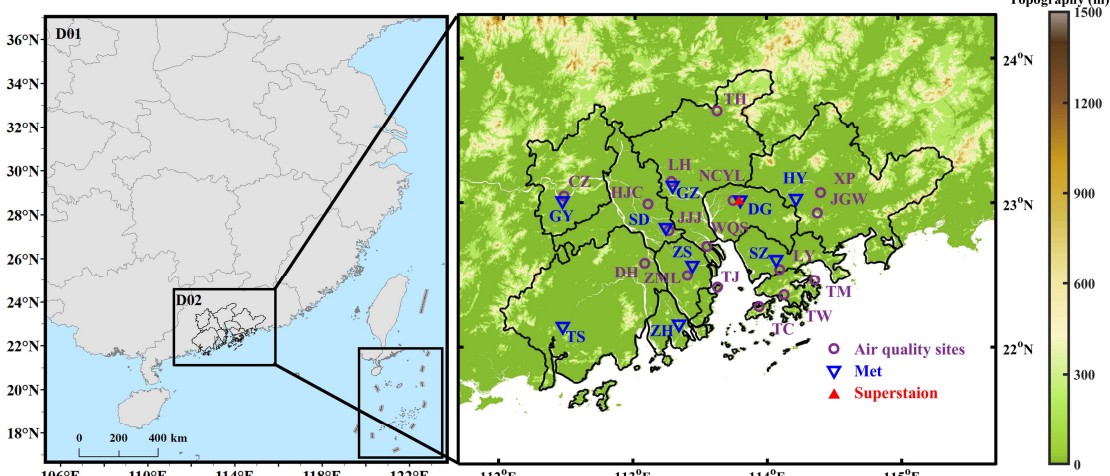


**Figure 1. Model domains and locations of 16 air quality monitoring sites (purple dots), 9 meteorological**
**stations (blue triangles), and the Dongguan superstation (red triangles). The figure on the right shows the**
**elevation of the terrain (m).**

152                                      **Table 1. Summary of the dataset used in this study**

| Description | Period | Sites | Temporal resolution | Spatial resolution | Purpose |
|---|---|---|---|---|---|
| Observed $O_3$ data | 2006-2019 | 16 sites | 1 h | - | Spatiotemporal analysis of NOI and NOP, model performance |
| Observed vertical $O_3$ data | 2019 | Dongguan superstation | 12 min | - | Analysis of an NOI event caused by Conv |
| Observed meteorological data | 2017.09.08-2017.09.15 | 9 sites | 1 h | - | Model performance |
| Observed Cloud-top Temperature (CTT) data | 2019 | Gridded data | 1 h | 0.1° | Indicator of the occurrence of convection |
| ERA5 reanalysis dataset | 2006-2019 | Gridded data | 1 h | 0.25° | Definition of LLJs and Conv |


**2.2 Definition of NOI and NOP**
For our analysis, we define a nocturnal $O_3$ increase (NOI) event as $O_3$ concentrations peaked at night
(from 21:00 LT to 06:00 LT the next day), with an increase in levels of at least 10 μg m$^{-3}$ compared to
the previous hour and a decrease of less than 10 μg m$^{-3}$ in the next hour. The corresponding nighttime
peak concentration of $O_3$ is referred to as the nocturnal $O_3$ peak (NOP) (Zhu et al., 2020). In this study,
based on the above observed hourly $O_3$ data at the 16 air quality monitoring sites, NOI events are
identified at each site, yet only one NOI event is recorded per night, regardless of how many NOI events
occur in a single night. In addition, the regional values of NOI and NOP from the 16 air quality
monitoring sites were averaged.
**2.3 Definition of LLJs and Conv**
Low-level jets (LLJs) and convective storms (Conv) are defined in this study based on the above site-
specific ERA5 reanalysis dataset. According to Banta et al. (2002) and Hodges and Pu (2019), LLJs are
defined as when vertical wind speed maxima occur below 800hPa and exhibit a decrease of at least 1.5
m s$^{-1}$ at vertical levels both above and below the levels of the maxima. It's worth noting that the LLJs
defined in this study only consider the turbulence mixing induced by their vertical wind shear.
Conv is defined by the following criterion: the mean K index (KI) is greater than 30 ℃ within 3 hours
prior to an NOI event (George, 1960; Johnson, 1982). The KI is calculated as follows:
$$KI = (T_{850} - T_{500}) + Td_{850} - (T_{700} - Td_{700}) \tag{1}$$

where $T_{850}$, $T_{700}$, and $T_{500}$ are the temperature (℃) at 850 hPa, 700 hPa, and 500 hPa, respectively,
and $Td_{850}$ and $Td_{700}$ are the dew point temperature at 850 hPa and 700 hPa, respectively.
Cloud-top temperature (CTT) was also introduced as an indicator of the occurrence of convective systems
and further used to evaluate the applicability of KI. The lower the CTT, the higher the probability of
convection event. According to the work of Ai et al. (2016), CTT lower than -35 ℃ indicates the
occurrence of convection. We randomly selected 10 nights with KI > 30 ℃ (Table S3) and 10 nights
with KI < 30 ℃ (Table S4) and examined the corresponding CTT values. In the cases with KI > 30 ℃,
the CTT values were lower than -35 ℃ in 10 out of 10 nights (Table S3). And the spatial distribution of
CTT showed that they had a distinct circular area with lower value over the selected sites, indicating the
occurrence of convective systems (Fig. S1). For the cases with KI < 30 ℃, 6 out of 10 nights were with
CTT higher than -35 ℃, while the rest 4 nights had no CTT data due to cloudless weather (Table S4).
The spatial distribution of CTT did not show the features of a convective system (Fig. S2), suggesting
that convection was not observed for the selected 10 cases with KI < 30 ℃. The above results suggest
that the KI > 30 ℃ criterion is a valid metric to capture the occurrence of convection.
In this study, an NOI event at each air quality site was classified into four categories: caused by LLJs
only, caused by Conv only, caused by LLJs and Conv (LLJs+Conv) at the same time, and caused by
other factors.
**2.4 Trend analysis**
In this study, the nonparametric Mann-Kendall (M-K) test (Mann, 1945) is used to determine the
statistical significance (p values) associated with the annual trends of NOI, NOP, MDA8 $O_3$, LLJs and
Conv, etc. A significance level of $p < 0.05$ was used to test the significance of the inter-annual trend. The
magnitude of a given trend is calculated by the nonparametric Theil-Sen (T-S) estimator (Sen, 1968).
The advantage of the M-K test and the T-S estimator is that they do not require prior assumptions of the
statistical distribution for the data and are resistant to outliers. The M-K test and the T-S estimator have
been widely used in previous $O_3$ trend studies (Wang, et al., 2019; Lu et al., 2020; Li et al., 2022).
**2.5 WRF-CMAQ model configuration**
Due to the lack of observed vertical profiles of wind speed, the WRF-CMAQ model is employed to
investigate the effects of LLJs on a selected NOI event. The NOI event induced by LLJs that occurred at
the Nancheng Yuanling (NCYL) site in Dongguan on September 13-14, 2017 is selected as a typical case.
The simulation was conducted during September 6-14 by using the WRF-CMAQ-IPR model with first
2 days used as model spin-up to eliminate the impact of IC (Jiménez et al., 2007).
The Weather Research and Forecasting model (WRFv3.9.1) is used to provide meteorological inputs to
drive the Community Multiscale Air Quality (CMAQ v5.3.1) model. The initial meteorological
conditions (IC) and boundary conditions (BC) are provided by the National Centers for Environmental
Prediction (NCEP) Final Analyses (FNL) dataset, with a spatial resolution of $1° × 1°$ and a temporal
resolution of 6 h. The main physics options used for the WRF model are shown in Table 2. Two-nested
domains are used in the WRF simulations, with 38 vertical layers from the surface to 100 hPa. Figure 1
shows the two nested modeling domains, with spatial resolutions of 27 km × 27 km and 3 km × 3 km for
the coarse (D01) and inner (D02) domains, respectively. D01 covers most regions of China and D02
covers the whole PRD region.
The CMAQ model is used to simulate the $O_3$ concentrations in the PRD region. The SAPRC07 and
AERO6 aerosol modules are used for gas-phase and particulate matter chemical mechanisms,
respectively (Carter, 2010; Wyat Appel et al., 2018). The chemical IC and BC for D01 are derived from
a global chemical transport model, the Model for Ozone and Related chemical Tracers, version 4
(MOZART4) (Emmons et al., 2010), and those for D02 are provided by the simulated results from D01.
The anthropogenic emissions for D01 are based on the 2016 Multi-resolution Emission Inventory for
China (MEIC), which has a grid resolution of 0.25° × 0.25° (Zheng et al., 2018). Those used for D02 are
based on the 2017 high-resolution emission inventory of the PRD region with a grid resolution of 3 km
× 3 km (Zhong et al., 2018), which includes the emission sectors of agriculture, biomass combustion,
incineration, dust, industrial processes, nonroad, solvent, storage, transportation, and waste disposal.
Biogenic emissions are calculated using the Model of Emissions of Gases and Aerosols from Nature
(MEGAN) v2.1 that was integrated into the CMAQ model (Guenther et al., 2006; Wang et al., 2011).
**Table 2. Model configurations**

| Model | Physical process | Parameterization scheme | Reference |
|-------|------------------|-------------------------|-----------|
| WRF | Microphysics | Lin | Lin et al. (1983) |
| | Longwave radiation | RRTMG | Iacono et al. (2008) |
| | Shortwave radiation | RRTMG | Iacono et al. (2008) |
| | Surface layer | Monin-Obukhov | Monin and Obukhov (1954) |
| | Planetary boundary layer | MYJ | Nakanishi and Niino (2006) |
| | Cumulus parameterization | Grell-3 | Grell and Dévényi (2002) |
| | Land surface | Noah land-surface model | Chen and Dudhia (2001) |
| CMAQ | Gas-phase chemistry | SAPRC 07 | Carter (2010) |
| | Aerosol chemistry | AERO6 | Carlton et al. (2010) |


In order to interpret the underlying atmospheric mechanisms for NOI events, the Integrated Process Rates
(IPR) analysis tool embedded in the WRF-CMAQ model is used to identify and quantify the contribution
of various physical and chemical processes to $O_3$. The processes include horizontal transport (HTRA),
vertical transport (VTRA), gas-phase chemistry (CHEM), dry deposition (DDEP), and cloud processes
(CLDS). Horizontal transport is the sum of horizontal advection and diffusion, and vertical transport is
the sum of vertical advection and diffusion. More details on the IPR analysis tool can be found in previous
work (Liu et al., 2010; Wang et al., 2010).
**2.6 Model evaluation**
The WRF-CMAQ simulation results are evaluated by comparison with available ground-based observed
$O_3$ and meteorological data. Statistical metrics including mean value ($\overline{Obs}$ and $\overline{Sim}$), mean bias (MB),
normalized mean bias (NMB), normalized mean error (NME), root mean square error (RMSE),
correlation coefficient (r), and index of agreement (IoA), are calculated as follows to evaluate model
performance.
$$MB = \overline{Obs} - \overline{Sim} \tag{1}$$

$$NMB = \frac{\sum_{i=1}^{n}(Sim_i - Obs_i)}{\sum_{i=1}^{n} Obs_i} \times 100 \tag{2}$$

$$NME = \frac{\sum_{i=1}^{n}|Sim_i - Obs_i|}{\sum_{i=1}^{n} Obs_i} \times 100 \tag{3}$$

$$RMSE = \sqrt{\frac{1}{n}\sum_{i=1}^{n}(Sim_i - Obs_i)^2} \tag{4}$$

$$r = \frac{\sum_{i=1}^{n}(Sim_i - \overline{Sim})(Obs_i - \overline{Obs})}{\sqrt{\sum_{i=1}^{n}(Sim_i - \overline{Sim})^2 \sum_{i=1}^{n}(Obs_i - \overline{Obs})^2}} \tag{5}$$

$$IoA = 1 - \frac{\sum_{i=1}^{n}(Sim_i - Obs_i)^2}{\sum_{i=1}^{n}\left(|Sim_i - \overline{Obs}| + |Obs_i - \overline{Obs}|\right)^2} \tag{6}$$

The evaluation protocols of the U.S. Environmental Protection Agency (EPA, 2017) are used to evaluate the performance of the meteorological parameters. The simulated results were accepted when the statistics met the criteria listed as follows: MB ≤ ±0.5 °C and IoA ≥ 0.8 for simulated T2; MB ≤ ±5% and IoA ≥ 0.6 for simulated RH; and MB ≤ ±0.5 m/s, RMSE ≤ 2.0 m/s, and IoA ≥ 0.6 for simulated WS10. The evaluation protocols of the Ministry of Environmental Protection of China (MEP, 2015) are used to evaluate the performance of $O_3$ and the simulated results were acceptable if the statistics met the criteria listed below: −15% < NMB < 15%, NME < 35%, and r > 0.4.

**3 Results and discussion**

**3.1 General characteristics of NOI events**

The average annual frequency of NOI events in the 16 sites across the whole PRD region from 2006 to 2019 is estimated to be $53 \pm 16$ d $yr^{-1}$, with an average annual NOP concentration of $58 \pm 11$ μg $m^{-3}$. LLJs are the primary factor causing NOI events, accounting for about 61%, followed by the combination of LLJs and Conv (LLJs+Conv) with a value of 16%, while the corresponding value is 11% for Conv (Fig. 2). The remaining 12% of NOI events that cannot be explained by LLJs and Conv may be related to other meteorological processes, such as mountain-valley breezes and sea-land breezes (Sousa et al., 2011; He et al., 2021).

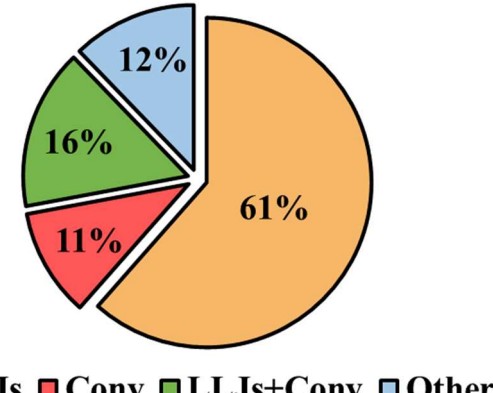


**Figure 2. The average relative contribution of different meteorological processes to NOI events during 2006-**
**2019.**
**3.2 Long-term trends of NOI events**
As depicted in Fig. 3a, the regional-average annual frequency of NOI events increases from $38 \pm 18$ d
yr$^{-1}$ in 2006 to as high as $67 \pm 18$ d yr$^{-1}$ in 2011 at a rate of 4.70 d yr$^{-1}$ (p<0.01) and gradually decreases
after 2012 at a rate of -0.72 d yr$^{-1}$ (p<0.05). A similar annual trend is observed for the frequency of total
downdrafts (sum of LLJs, LLJs+Conv, and Conv) (Fig. 3b). The frequency of total downdrafts increased
at a rate of 4.02 d yr$^{-1}$ (p<0.01) before 2012 and decreased at a rate of -0.53 d yr$^{-1}$ (p<0.05) thereafter,
which is significantly positively correlated with NOI events, with a Pearson correlation coefficient (r) of
0.96 (p<0.01). Among the total downdrafts, LLJs exhibit a similar pattern with NOI events (r=0.89,
p<0.01), further suggesting that LLJs are the predominant driver. Conv presents a continuously
increasing trend over the whole 14-year period, with a rate of 0.26 d yr$^{-1}$ (p<0.01), and the frequency of
LLJs+Conv does not show obvious variation.
Both the frequency of NOI and LLJs present increasing trends before 2012 and decreasing trends
thereafter, which was related to urbanization. Previous studies have shown that urbanization has large
effects on the frequency of LLJs by changing surface conditions (roughness and soil moisture) and further
affecting the turbulence and geostrophic wind speed (McCorcle, 1988; Fast and McCorcle, 1990;
Kallistratova, 2008; Nikolic et al., 2019; Ziemann et al., 2019). Kallistratova (2008) and Nikolic et al.
(2019) pointed out that negative correlation was found between urban areas and the frequency of LLJs.
During 1987-2017, the urban areas in the PRD region grew at an average rate of 8.82% yr$^{-1}$ (Yang et al.,
2019a) and reached maximum urban land expansion growth rate of 6.66% during 2010-2015 (Zhang et

al., 2021). Therefore, the trends for the frequency of NOI and LLJs were quite different during these two periods (2006-2011 and 2012-2019).

Although the percentage of NOI events caused by Conv alone is relatively small compared to those caused by LLJs (Fig. 2), it is noteworthy that the frequency of Conv-induced NOI events was on the increase during the 14-year period (Fig. 3b), which is also mainly related to the rapid urbanization in the PRD region in recent years (Yang et al., 2019a; Zhang et al., 2021). Surface roughness increase due to city expansion led to the greater frequency and intensity of convection in the form of enhanced mechanical turbulence (Li et al., 2021), thus an increase in the frequency of Conv-induced NOI events. The role of Conv in the occurrence of NOI events is expected to amplify in the future if the urbanization trend in China continues (Seto et al., 2012; Marelle et al., 2020).

In contrast to the annual trend of NOI frequency, the nocturnal $O_3$ peak (NOP) value shows an upward trend during 2006-2019, with a slower growth rate of 0.54 $\mu g\ m^{-3}\ yr^{-1}$ (p<0.05) before 2015 and a faster growth rate of 4.76 $\mu g\ m^{-3}\ yr^{-1}$ (p<0.01) thereafter (Fig. 3c). The maximum daily 8-h average (MDA8) $O_3$ mixing ratio exhibits a similar pattern to NOP, with an increase rate of 0.39 $\mu g\ m^{-3}\ yr^{-1}$ (p<0.05) before 2015 and 9.21 $\mu g\ m^{-3}\ yr^{-1}$ (p<0.01) thereafter (Fig. 3d). NOP is significantly positively correlated with MDA8 $O_3$, with r up to 0.88 (p<0.01). This implies that daytime $O_3$ concentration levels potentially affect NOP concentrations. The variations of NOP and MDA8 $O_3$ during the two periods (2006-2015 and 2016-2019) are more likely related to the change in precursor emissions. The continuous increase in the emissions of anthropogenic VOCs and $NO_x$ resulted in the gradual increase of $O_3$ concentrations between 2006 and 2012 (Ma et al., 2016; Li et al., 2017; Zhong et al., 2018; Liao et al., 2021). However, since the implementation of Air Pollution Prevention and Control Action Plan (APPCAP) in 2013, $NO_x$ emissions was dramatically decreased by 21% in 2017 compared to 2013 (Feng et al., 2019; Yang et al., 2019b). The weakening of NO titration caused by the dramatic decrease in $NO_x$ emissions and the continuously increasing VOCs emissions due to the lack of controls became important drivers of the sharp rise in $O_3$ since 2015 (Li et al., 2019; Mousavinezhad et al., 2021; Li et al., 2022). Furthermore, the decreasing $PM_{2.5}$ levels and the increasing atmospheric oxidizing capacity in the PRD region in recent years have also been considered as important contributors to accelerated $O_3$ growth during 2016-2019 (Gong et al., 2018; Li et al., 2019; Han et al., 2019). Consequently, NOP and MDA8 $O_3$ present slower increase rate before 2015 and higher increase rate thereafter.

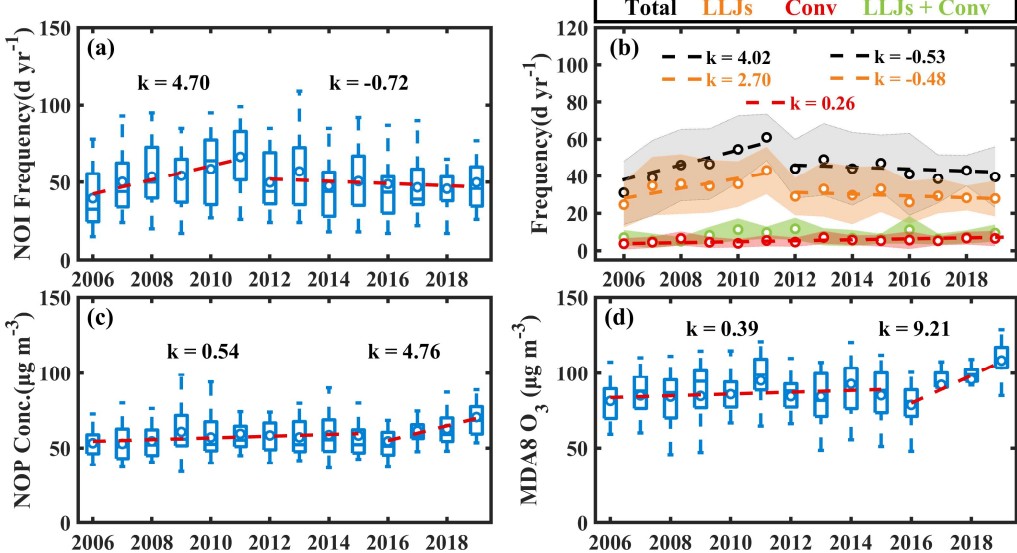


**Figure 3. Regional-average annual trends of (a) frequency of NOI events, (b) frequency of total downdrafts**
**(black), LLJs (orange), LLJs+Conv (green), and Conv (red) that can induce NOI events, (c) NOP**
**concentrations, and (d) MDA8 $O_3$ concentrations in the PRD region during 2006-2019. The units of k (Sen's**
**Slope) are $d^{-1}$ $yr^{-1}$ in (a) - (b) and $\mu g$ $m^{-3}$ $yr^{-1}$ in (c) - (d). Linear trends significant at the 95% confidence level**
**are illustrated with dashed lines. The error bars indicate the range of deviations for the 16 air quality sites.**
**3.3 Seasonal variations of NOI events**
NOI events exhibit obvious seasonal variation (Fig. 4a), with relatively higher frequency observed in
spring ($18 \pm 4$ d $yr^{-1}$) and autumn ($20 \pm 5$ d $yr^{-1}$) and lower frequency in summer ($14 \pm 3$ d $yr^{-1}$) and winter
($16 \pm 3$ d $yr^{-1}$). LLJs are the dominant inducer of NOI events in spring, autumn, and winter (Fig. 4b),
while in summer, the dominant factors are LLJs+Conv and Conv because convective activity is more
intense during summer (Chen et al., 2014). Given that LLJs can enhance turbulence below the jet and
create favorable formation conditions for Conv (Trier et al., 2017; Du and Chen, 2019), most Conv events
preferentially occur on days when LLJs exist in the PRD region (Chen et al., 2014), which makes
LLJs+Conv the main contributor in summer.
In terms of NOP (Fig. 4c), relatively higher concentrations are observed in spring and autumn, with
values of $59 \pm 12$ $\mu g$ $m^{-3}$ and $66 \pm 10$ $\mu g$ $m^{-3}$, respectively, while the concentration in summer is the
lowest ($44 \pm 7$ $\mu g$ $m^{-3}$). The MDA8 $O_3$ has a similar seasonal variation to NOP except in winter (Fig. 4d);
it is high in spring ($111 \pm 15$ $\mu g$ $m^{-3}$) and autumn ($120 \pm 13$ $\mu g$ $m^{-3}$) and low in summer ($88 \pm 15$ $\mu g$ $m^{-3}$).
In winter, surface MDA8 $O_3$ was the lowest ($86 \pm 12$ $\mu g$ $m^{-3}$) while NOP remained at relatively high
levels ($56 \pm 10$ $\mu g$ $m^{-3}$). This is because the higher $O_3$ concentrations in the lower troposphere in winter
allow more $O_3$ to be transported downward during the NOI period, resulting in a higher NOP
concentration in winter, as shown by the seasonally observed vertical $O_3$ profile at the Dongguan
superstation (Fig. 5). As illustrated in Fig. 5, higher $O_3$ concentrations are observed at 200 to 750 m
altitude in winter than in summer. A similar result was also observed in Hong Kong (Liao et al., 2021).
This is mainly due to the typical Asian monsoon circulation, which brings clean marine air to the lower
troposphere of the PRD region in summer and dilutes polluted air masses inland, while it brings pollutant-
laden air from mainland China in winter resulting in higher $O_3$ concentrations over the PRD region (Wang
et al., 2009).

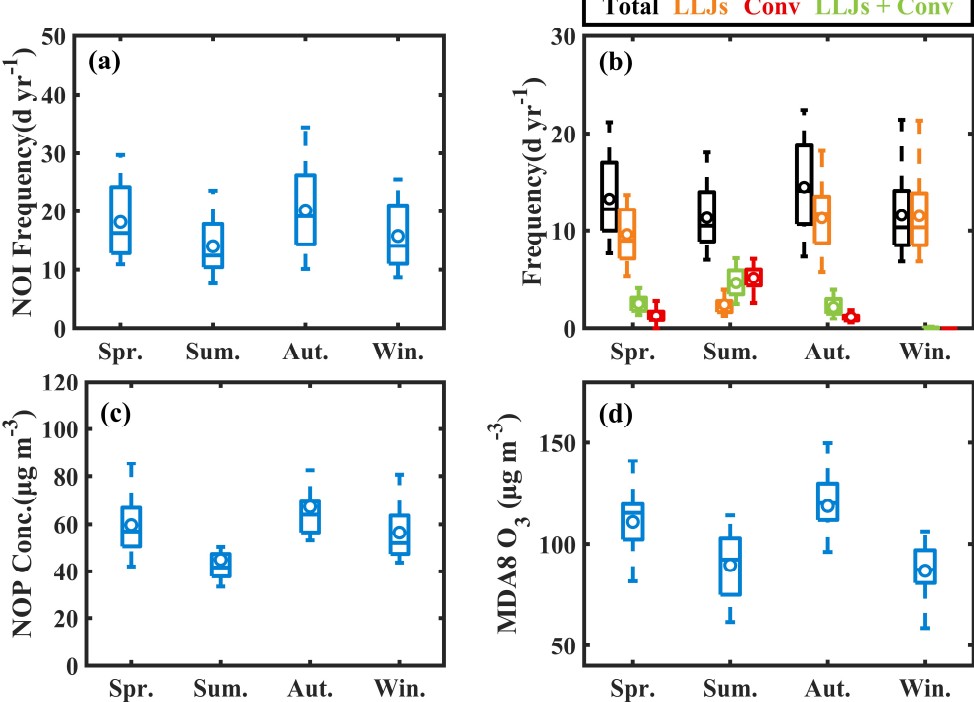


**Figure 4. Seasonal variation of (a) frequency of NOI events, (b) frequency of total downdrafts (black), LLJs**
**(orange), LLJs+Conv (green), and Conv (red) that can induce NOI events, (c) NOP concentrations, and (d)**
**MDA8 $O_3$ concentrations in the PRD region during 2006-2019. The error bars indicate the range of deviations**
**for the 16 air quality sites.**

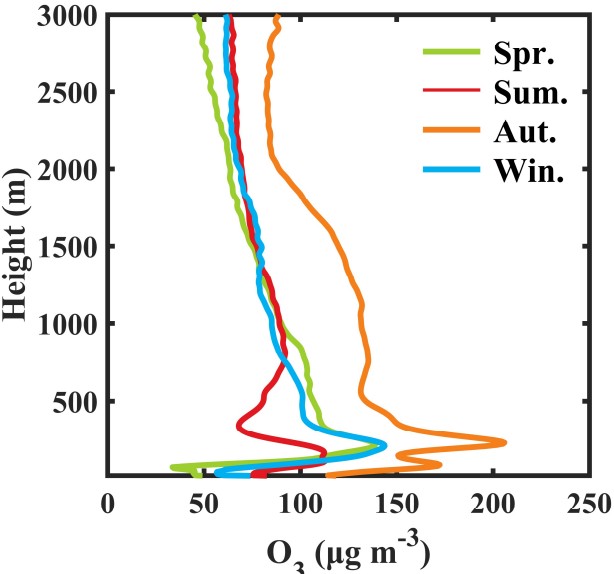

**Figure 5. Seasonally averaged vertical distribution of O₃ concentrations at the Dongguan superstation from the surface to an altitude of 3000 m in 2019. Spring (Spr.): March-May; Summer (Sum.): June-August; Autumn (Aut.): September-November; Winter (Win.): December-February.**

**3.4 Diurnal variation of NOI events**

Distinct diurnal variation is observed in NOI events (Fig. 6a) with an increasing trend from 21:00 to 03:00 and a decreasing trend thereafter. It is estimated that about 60% of the events occurred in the middle of the night (11:00-03:00). The LLJs that can induce NOI events show a similar diurnal variation to the frequency of NOI events and often occur around at midnight. However, the frequency of total LLJs differs from the frequency of LLJs that can induce NOI events (Fig. 6b), as it increases steadily from 21:00 to 00:00 and remains stable after 00:00. This suggests that LLJs are not the only factor that determines whether an NOI event can develop, and O₃ concentration in the RL can also affect the development of an NOI event. As the sun sets and the daytime boundary layer fades away, the O₃ produced during daytime remains at a relatively high level in the RL during 21:00-03:00. During this period, the occurrence of LLJs tends to increase the probability of NOI events. After 03:00, the O₃ concentrations in the RL decreased due to horizontal transport to downwind area and vertical transport (e.g., LLJs, convection, O₃ dry deposition process) during 21:00-03:00, which reduced the amount of O₃ that can be transported downward. Hence, even though the frequency of the total LLJs is relatively high after 03:00, the lower O₃ content in the RL results in less O₃ being transported downward to form an NOI event, which ultimately decreases the frequency of NOI events. As illustrated in Fig. 6c, the trends of NOP concentrations from 21:00 to 06:00 also reflect the fact that O₃ concentrations in the RL are higher

during 21:00-00:00 and lower during 00:00-06:00. Therefore, the development of an NOI event is

influenced by the combination of a downdraft induced by meteorological processes, and the level of $O_3$

concentrations.

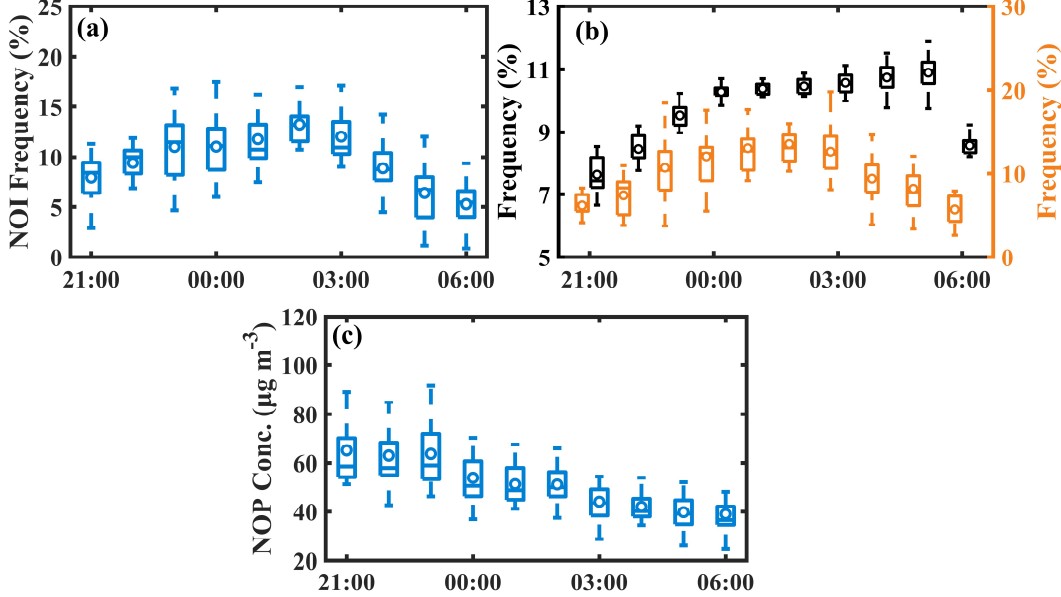

**Figure 6. Diurnal variation of (a) the frequency of NOI events, (b) frequency of total LLJs (black) and the LLJs that can induce NOI events (orange), (c) NOP concentrations during 21:00-06:00 (LTC) in the PRD region during 2006-2019. The error bars indicate the range of deviations for the 16 air quality sites.**

**3.5 Spatial distribution of NOI events**

As most NOI events are caused by LLJs, LLJs are taken as an example to explore the role of

meteorological processes in the spatial distribution of NOI events. The spatial distribution of the average

annual frequency of NOI events and LLJs in the PRD region from 2006 to 2019 is shown in Fig. 7a, and

the spatial distribution of MDA8 $O_3$ concentrations obtained by Kriging's interpolation method is shown

in Fig. 7b. Obvious geographical variations are observed for NOI events, with a higher frequency in the

eastern PRD region, coupled with a higher frequency of LLJs, although the MDA8 $O_3$ concentrations are

relatively lower in these regions. In the central PRD region, despite the highest MDA8 $O_3$ concentrations,

the frequency of NOI events was the lowest, implying a more important role of vertical transport induced

by meteorological processes in the formation of NOI events. At the three sites located in the southern

part of the PRD regions (TJ, TW, and TC), the frequency of NOI events was the highest while the

frequency of LLJs was not. This is because these three sites were also affected by non-LLJs

(=Conv+(LLJs+Conv)+Other) processes with comparable contributions of LLJs and non-LLJs to the

NOI events. And the contributions of LLJs (60-70%) were higher than those of non-LLJs at the rest of
sites (Table S5).

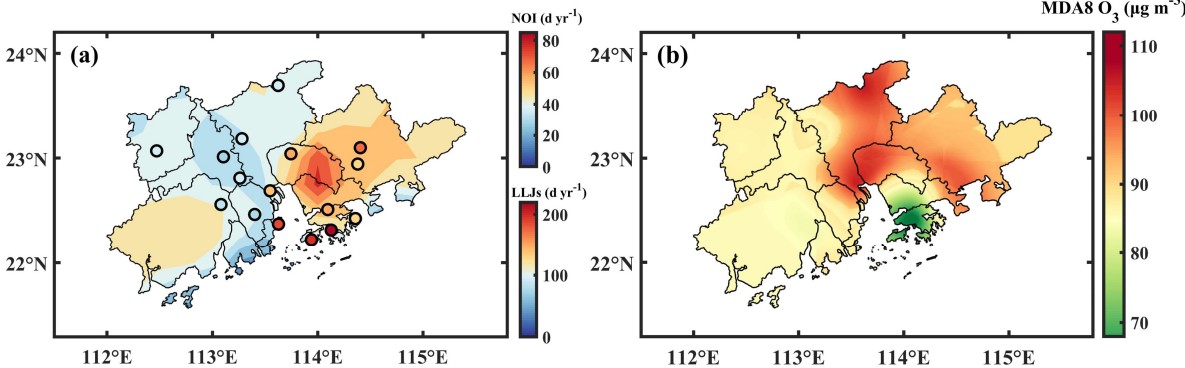

**Figure 7. Spatial distribution of annual average (a) NOI event frequency (points) and LLJs frequency**
**(contours), (b) MDA8 $O_3$ concentrations.**

### 3.6 Causative analysis of NOI events: Convective storms trigger

In order to elaborate the underlying atmospheric mechanisms for Conv-induced NOI events, a distinct
NOI event associated with Conv observed at the Nancheng Yuanling (NCYL) site in Dongguan on
September 3-4, 2019 is taken as a typical example for further discussion. The vertical $O_3$ profile data
observed at the Dongguan superstation are used to represent the vertical $O_3$ distribution at the NCYL site
during this NOI event, since the distance between these two stations is only 3 km.
The KI remains above 36 ℃ (Fig. 8a) and the vertical velocity show continuous updraft trends at 1-3 km
altitude from 14:00 to 23:00 (Fig. 8b), indicating a high possibility of convection. Although the
magnitude of vertical velocity was relatively low, it also has been found in previous studies (Ploeger et
al., 2021). In addition, the spatial distribution of CTT show that the CTT value at 18:00 over Dongguan
was around -70 ℃ (Fig. 9a), which was lower than the criterion (-35 ℃) for the happening of convection
process. The results of KI, the vertical velocity and the CTT indicate that the possibility of a convection
process is high. Thus, the precipitation occurs during 16:00-20:00 was a convective precipitation. The
effect of rainfall on $O_3$ removal is relatively small after sunset, because wet deposition of $O_3$ occurs
through the removal of the precursors $HNO_3$ and $H_2O_2$ by water vapor under solar radiation, which is
indirect and rather peripheral, and the effect of heterogeneous processes on $O_3$ removal is weak (Jacob,
2000; Awang et al., 2015; Zhu et al., 2020). Therefore, the unconsumed $O_3$ remains stable in the RL. As
illustrated in Fig. 8c, higher $O_3$ concentrations are found in the RL after 18:00, reaching around 200 μg
$m^{-3}$. At 21:00, a strong updraft suddenly appears above 1.5 km (Fig. 8b) and a distinct area of low CTT
values (around -66 °C) could be observed over Dongguan (Fig. 9b), confirming the happening of updraft.
The updraft subsequently caused a strong compensating downdraft below 1 km at 22:00-23:00 (Fig. 8b).
The downdraft then breaks through the stable nocturnal boundary layer and transports $O_3$ from the RL to
the surface (Fig. 8c). Hence, an NOI event occurs at 23:00, with $O_3$ concentrations increased from 45 μg
$m^{-3}$ at 22:00 to 59 μg $m^{-3}$ at 23:00. Although the modeled downdraft occurred at 22:00-23:00 (Fig. 8b)
was around half an hour later than the observed $O_3$ intrusion into the nocturnal boundary layer (Fig. 8c)
due to the model bias, the modeled results can still generally capture the occurrence of convection
processes.

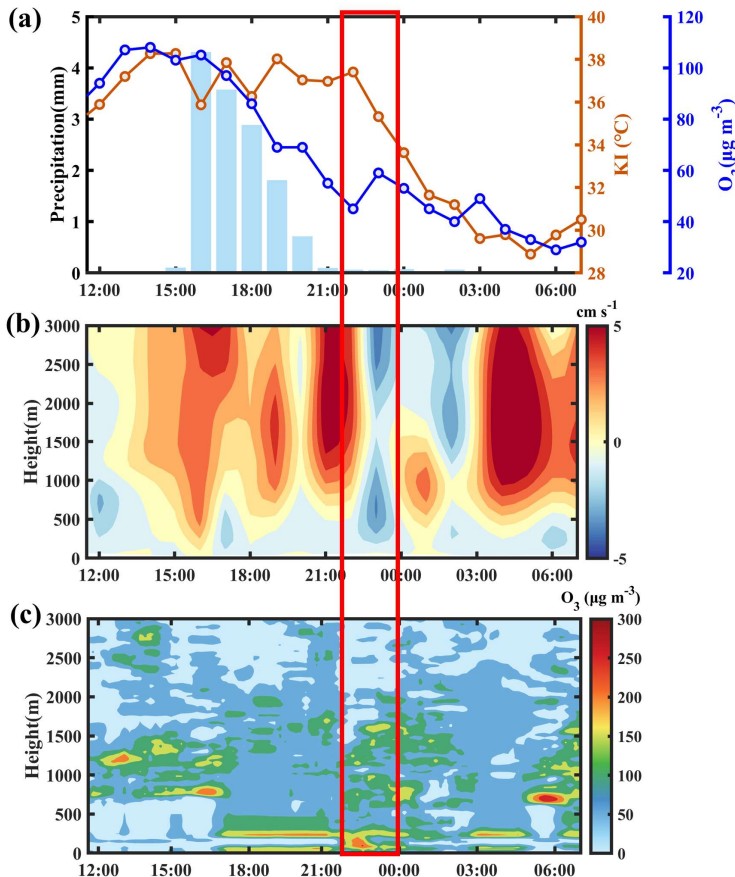


**Figure 8. (a) Hourly variations of KI (brown line), $O_3$ concentrations (blue line), and hourly precipitation**
**amount (blue bar), (b) vertical wind velocity, with positive and negative values related to updrafts and**
**downdrafts, (c) vertical profile of $O_3$ concentrations at the NCYL site in Dongguan on September 3-4, 2019.**

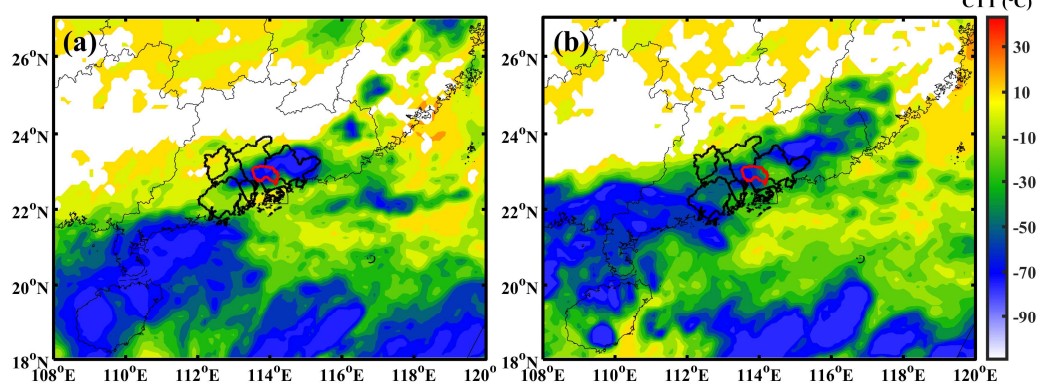


**Figure 9. Spatial distribution of cloud-top temperature (CTT) at (a) 18:00 and (b) 21:00 LT on September 3,**

**2019.**

**3.7 Causative analysis of NOI events: LLJs trigger**

Another typical NOI event induced by LLJs occurred at the NCYL site in Dongguan on September 13-

14, 2017 was simulated using the WRF-CMAQ-IPR model because no vertical profiles of wind speed

were observed. Model performance was first evaluated. Comparisons of the simulated meteorological

parameters with observations for the 9 sites in the PRD region during September 8-14, 2017 are shown

in Fig. S3, with statistical indices reported in Table S6. The results show that WS10 was reasonably well

simulated, as the regional average of MB, RMSE, and IoA met the EPA criteria mentioned in section 2.6.

The simulated regional average of RH and T2 were slightly overestimated (MB=-6.1%) and

underestimated (MB=1.4 °C), respectively, while they performed well at the Dongguan site, where both

the MB (RH=-1.0%, T2=0.5 °C) and IoA (both are 1.0) met the EPA criteria. The MB of the simulated

WS10 at the Dongguan site was slightly underestimated (MB=0.7 m s$^{-1}$), with the RMSE and IoA met

the EPA criteria.

Comparisons of simulated O$_3$ with hourly observations during September 8-14, 2017 are shown in Fig.

S3, with statistical indices reported in Table S7. The simulated O$_3$ showed a good performance in the

PRD region, with NMB (-4.8%), NME (17.7%) and r (1.0) all met the MEP criteria, while it was slightly

underestimated at the NCYL site in Dongguan but still met the MEP criteria (NMB=-12.7%,

NME=29.8%, r=0.8). Therefore, the simulation results of the meteorological parameters and O$_3$ are

reasonable and reliable for further analysis.

Figure 10a shows the time series of simulated O$_3$ concentrations and the contributions of different

processes to surface O$_3$ concentrations at the NCYL site in Dongguan during September 13-14, 2017.

During the night on September 13, the $O_3$ concentration increased from 45 μg m$^{-3}$ at 21:00, reached the
peak of 85 μg m$^{-3}$ around 22:00 and dropped to 15 μg m$^{-3}$ at 00:00. Before 21:00, the magnitude of the
negative contribution of chemical processes to $O_3$ was greater than the positive contribution of vertical
transport, resulting in net $O_3$ depletion. This suggests that gas-phase chemistry processes such as NO
titration are the main pathway for $O_3$ loss at night. At 21:00, the vertical and horizontal transport
contribution increased abruptly by 48 μg m$^{-3}$ and 27 μg m$^{-3}$, respectively, while the chemical depletion
remained constant. At this point, the net $O_3$ concentration turned from loss to production (51 μg m$^{-3}$). In
terms of vertical distribution (Fig. 10b), a positive contribution of both vertical and horizontal transport
can be found at the surface, while vertical transport became negative in the upper layers and horizontal
transport remained positive, indicating the occurrence of a downdraft. In addition, the wind profile
showed a typical LLJs characteristic (Fig. 10b), with a maximum wind speed of about 12 m s$^{-1}$ at 1 km
altitude and a wind speed difference of more than 3 m s$^{-1}$ above and below. Figure 11 further presents
the process of vertical transport during an NOI event. Compared to normal days, the nocturnal boundary
layer during the NOI event was more unstable and turbulent, with significant upward and downward
transport. At around 1 km, there was a straight stream over the NCYL site during the NOI event (Fig.
11b). This suggested that LLJs broke the stable structure between the nocturnal boundary layer and the
RL and enhanced the strength of turbulence (Caputi et al., 2019). The LLJs-induced turbulence promoted
mixing between the upper and lower layers and continuously transported $O_3$ from the upper layer to the
surface, causing an unusual surge in $O_3$ at the surface and leading to an NOI event. As a result, the LLJs
process contributed as much as 40 μg m$^{-3}$ $O_3$ from the upper layer to the surface during this NOI event.

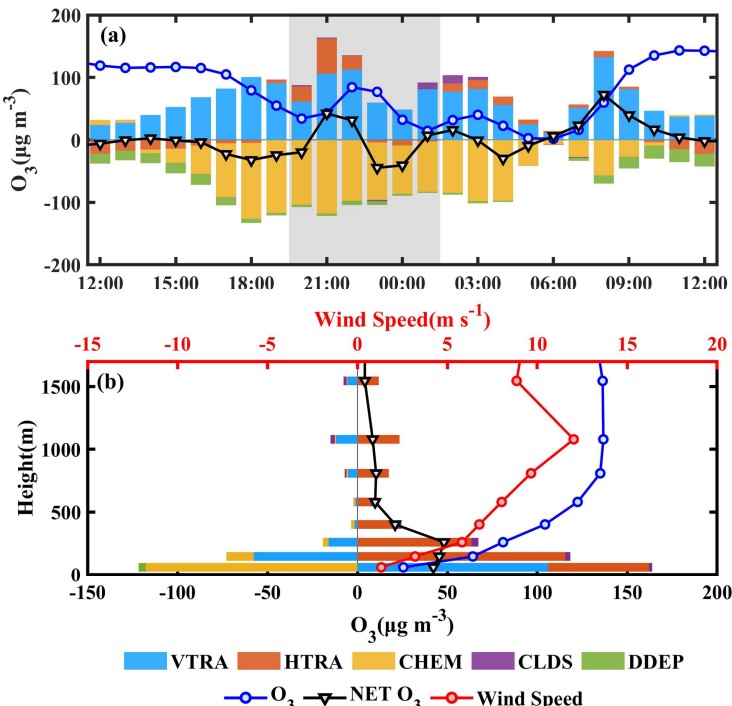


**Figure 10. Contribution of individual processes to (a) hourly O₃ concentration near the surface during**
**September 13-14, 2017 and (b) vertical O₃ concentration at 21:00 on September 13, 2017. VTRA: vertical**
**transport, the net effect of vertical advection and diffusion; HTRA: horizontal transport, the net effect of**
**horizontal advection and diffusion; CHEM: gas-phase chemistry; CLDS: cloud processes; DDEP: dry**
**deposition; NET: the net change in O₃ due to all atmospheric processes.**

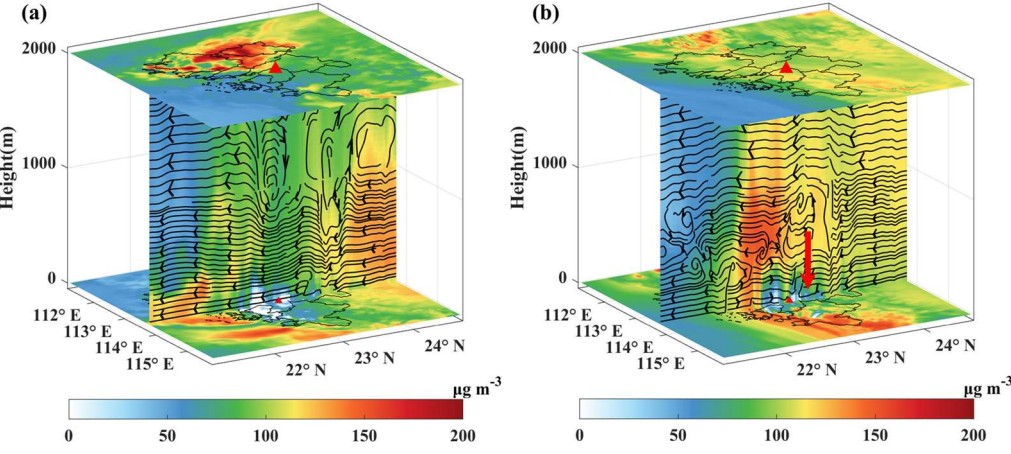


**Figure 11. Vertical profiles of O₃ concentrations at 21:00 during (a) a normal day (September 12, 2017) and**
**(b) an NOI event (13 September 2017). Red triangles represent the NCYL site in Dongguan, contours**
**represent O₃ concentrations (μg m⁻³), and black lines and arrows indicate vertical airflow and its direction.**
**3.8 Comparison and Prospects**
Zhu et al. (2020) identified an NOI event frequency of 16-19 d yr⁻¹ in the summers from 2014 to 2015 in
Beijing, China, with nocturnal O₃ maxima ranging from 45 to 85 μg m⁻³, which is comparable to our
result of NOI frequency ($14 \pm 3$ d yr$^{-1}$) and slightly higher than our NOP concentration ($44 \pm 7$ μg m$^{-3}$)
in summer. Sousa et al. (2011) analyzed nocturnal $O_3$ maxima events (maxima higher than the average
nocturnal $O_3$ concentration of 10 μg m$^{-3}$) during 2005-2007 in northern Portugal and found that the
frequencies of nocturnal $O_3$ maxima were between 40% and 50% in urban areas and 15% in rural areas,
which is higher than our NOI frequency (53 d yr$^{-1}$, 14.5%). Other studies focusing on short-term nocturnal
$O_3$ maxima cases found that NOP concentrations were 30-50 μg m$^{-3}$ in the Lower Fraser Valley, British
Columbia Canada (Salmond and McKendry, 2002), 20-60 μg m$^{-3}$ in Senegal (Grant et al., 2008), and 40-
80 μg m$^{-3}$ in North America (Kuang et al., 2011; Hu et al., 2013; Sullivan et al., 2017), values that are
comparable to our results ($58 \pm 11$ μg m$^{-3}$).
Our study emphasizes the importance of meteorological processes as well as daytime $O_3$ concentration
in the occurrence of NOI events, implying that higher NOP may occur during a severe daytime $O_3$
pollution period under the effect of meteorological processes. The occurrence of NOI events is likely to
impact the $O_3$ levels on the following day, which makes $O_3$ prevention more complex and challenging
(Ravishankara, 2009; Sullivan et al., 2017). However, the relationship between NOI events and the
following daytime $O_3$ pollution remains unclear and controversial. Kuang et al. (2011) and Sullivan et al.
(2017) revealed that NOI events led to a higher increasing rate of $O_3$ and worse air quality on the
following day, while Klein et al. (2019) and Caputi et al. (2019) observed lower $O_3$ levels during the
daytime following NOI events. To further explore the relationship between the daytime MDA8 $O_3$ and
nighttime NOP in the PRD region, we display the correlation between the MDA8 $O_3$ and the following
night's NOP (shorthand MDA8-NOP) (Fig. 12a) and the NOP and the following MDA8 $O_3$ (shorthand
NOP-MDA8) (Fig. 12b), respectively. The results show that MDA8 $O_3$ was positively correlated with
NOP with a correlation coefficient of 0.63 (p<0.01) and 0.56 (p<0.01) for MDA8-NOP and NOP-MDA8,
respectively, suggesting an interplay between daytime $O_3$ and NOP in the PRD region.

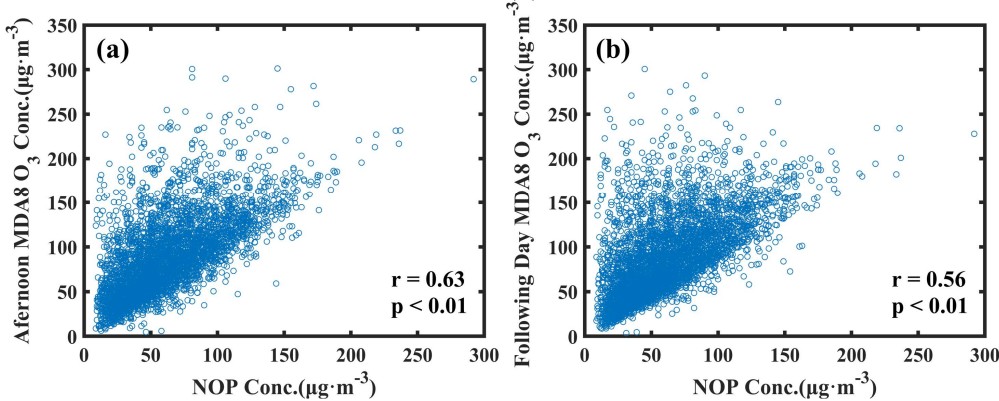


**Figure 12. Correlation between (a) the afternoon MDA8 O₃ concentration and the following night's NOP**
**concentration and (b) the NOP concentration and the following afternoon MDA8 O₃ concentrationn**
**4 Conclusion**
In this study, based on in-situ $O_3$ concentrations, observed vertical profiles of $O_3$, ERA5 reanalysis
datasets, and the WRF-CMAQ model, the spatial and temporal characteristics of NOI events are
comprehensively presented and the role of vertical transport in NOI events in the PRD region from 2006
to 2019 is further quantified.
The average annual frequency of NOI events is estimated to be $53 \pm 16$ d yr$^{-1}$ from 2006 to 2019, with
an annual average of $58 \pm 11$ μg m$^{-3}$ for nocturnal $O_3$ peak (NOP). LLJs are the dominant factors causing
NOI events (61%), followed by the combination of LLJs and Conv (LLJs+Conv) with a value of 16%.
The high correlation between NOI events and the frequency of LLJs in the annual trend (r=0.89, p<0.01)
supports the important influence of LLJs on the occurrence of NOI events. Although the contribution of
Conv to NOI events is relatively small, Conv-induced NOI events steadily increased at a rate of 0.26 d
yr$^{-1}$ during this 14-year period due to the impact of urbanization. Moreover, the significant positive
correlation between NOP and maximum daily 8-h average (MDA8) $O_3$ in annual (r=0.88, p<0.01) and
seasonal trends (r=0.80, p<0.01) and a higher NOI frequency (60%) during the first half of the night
imply that daytime $O_3$ concentrations are also an important factor influencing the formation of NOI
events.
Two typical NOI events caused by LLJs and Conv, respectively, further demonstrate that downdrafts
from enhanced turbulence are the direct cause of NOI events, as these can transport $O_3$ from the RL to
the surface. The difference is that LLJs induce downdrafts by a fast-moving air mass enhancing shear
below, whereas Conv induce downdraft by compensating downdrafts.
This study emphasizes the importance of vertical transport induced by LLJs and Conv, and daytime $O_3$
concentration in the formation of NOI events and highlights the key role of vertical transport in linking
daytime and nighttime $O_3$ pollution. This study not only provides a new perspective and better
understanding to reconceptualize the role of meteorology in daytime and nighttime $O_3$ pollution, but also
provides a reference for other regions with ground-level $O_3$ pollution.

*Data availability.*   In-situ hourly $O_3$ concentrations at 16 stations across the PRD region from 2006 to
2019 can be downloaded from http://113.108.142.147:20047; the observed hourly meteorological data
at the 9 sites across the PRD region can be downloaded from http://www.cma.gov.cn/; the ERA5
reanalysis dataset can be downloaded from https://cds.climate.copernicus.eu/cdsapp#!/home; and the
vertical $O_3$ profile data available upon request.

*Author contributions.*   YW and WC designed the research. YW did the data analysis and simulation
work and prepared the draft with support and editing from WC. YY and QX contributed to data analysis.
SJ and XW contributed to paper revision.

*Competing interests.*   The authors declare that they have no conflict of interest.

*Acknowledgements.*   This study was supported by the Key-Area Research and Development Program of
Guangdong Province(2020B1111360003), the National Natural Science Foundation of China (41905086,
42121004, 41905107, 42077205, 41425020), the National Key Research and Development Plan
(2019YFE0106300), the Special Fund Project for Science and Technology Innovation Strategy of
Guangdong Province (2019B121205004), the AirQuip (High-resolution Air Quality Information for
Policy) Project funded by the Research Council of Norway, the Collaborative Innovation Center of
Climate Change, Jiangsu Province, China, and the high-performance computing platform of Jinan
University.

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
