# Peer review of "Quantitative impacts of vertical transport on long-term"

_Atmospheric Chemistry and Physics, 2022_

## Author Comment (AC1)

**Responses to Reviewers' comments**

To the esteemed Editor and Reviewers,

We would like to thank the reviewers for the time and efforts in reviewing our manuscript. We have revised the manuscript according to the reviewers' detailed comments, which we sincerely hope the correction will meet with the high publishing standard of the journal. Please find the point-to-point responses to the reviewers' comments as follows:

Reviewers' comments are in black.

Author's responses are in blue color.

Changes in the manuscript are in red color.

Sincerely,

Weihua Chen

On behalf of the authors

**Response to Reviewer #1:**

**Referee #1 comment:**

Wu et al. provides a novel framework for capturing the essence of nocturnal ozone increases (NOI), which is an important area of research often neglected in analysis of the ozone budget. They break the causes of NOI down into clearly discernable phenomena and present evidence that the majority of NOI events in the Pearl River Delta (PRD) are caused by Low Level Jets (LLJ). Further, they present a detailed case study of a LLJ NOI event, as well as a convective storm (Conv) NOI event.

The framework is intriguing and presents a valuable contribution to the literature, however, some modifications should be made prior to publication. In particular, the authors need to more clearly define their methodology and make a stronger case for using the K index as a proxy for convective storm events.

Response:

We thank the reviewer for the positive comments. We have revised our paper according to your comments as follow:

(1) We have reorganized the **Data and methods** section and provided clearer description regarding methodology used in this study. Please refer to our detailed responses to comments #1, #2, #3, and #5.

(2) We have introduced an additional indicator (cloud-top temperature, CTT) to prove the applicability of K index. Please refer to our detailed responses to comments #5.

**General Comments:**

1.  The definition of NOI first appears in lines 118-120, where values increase by at least 10 μg m$^{-3}$. However, this comes from a 2020 reference and not every preceding study in the introduction that mentions NOI contains findings that are consistent with this strict definition (e.g. Caputi et al. 2019). It would be helpful if the authors clarify that (I assume) this is the definition they employ specifically in their study (e.g. "for our analysis, we define NOI as …"). Additionally, the definition needs to be clearer. For example, I am left unsure whether "values increasing by at least 10 μg m$^{-3}$" mean increasing from the daytime minimum, the previous hour's value, or what exactly.

Response:

According to the reviewer's comments, we have provided a specific definition of NOI used in this study in Lines 155-158:

'For our analysis, we define a nocturnal O$_3$ increase (NOI) event as O$_3$ concentrations peaked at night

(from 21:00 LT to 06:00 LT the next day), with an increase in levels of at least 10 μg m$^{-3}$ compared to the previous hour and a decrease of less than 10 μg m$^{-3}$ in the next hour. The corresponding nighttime peak concentration of O$_3$ is referred to as the nocturnal O$_3$ peak (NOP) (Zhu et al., 2020).'

2. Some aspects of the methodology need clarification. In Section 3.1, are the statistics (e.g. 53 +/- 16 d yr$^{-1}$) from an aggregate of the air quality monitoring stations, and are the error values and error bars in Figures 3 and 4 calculated by a pooled standard deviation? The authors then discuss the proportions of events attributable to LLJ vs. Conv, does this come from the ERA5 data? If so, are the causes of an individual NOI event (LLJ, Conv, LLJ+Conv, Other) determined by an instantaneous snapshot of the meteorological conditions over the air quality station, or a regional average for a given night? Please connect the dots between the different methods discussed (e.g. CMAQ, IPR, air quality stations, meteorological stations, ERA5) and where specifically they employed in the results.

Response:

(1) The statistic of average annual frequency of NOI events (53 ± 16 d yr$^{-1}$) is an average value of the 16 air quality monitoring stations. We have clarified it in Lines 161-162:

'In addition, the regional values of NOI and NOP from the 16 air quality monitoring sites were averaged.'

(2) The error values and error bars in Figures 3 and 4 indicate the range of deviations for the different station. We have clarified it in Line 317, Line 344 and Line 373:

'… The error bars indicate the range of deviations for the 16 air quality sites.'

(3) The proportions of NOI events attributable to LLJs vs. Conv are calculated based on the ERA5 data. We have clarified it in Lines 164-165:

'Low-level jets (LLJs) and convective storms (Conv) are defined in this study based on the above site-specific ERA5 reanalysis dataset.'

(4) The causes of an individual NOI event are determined by an instantaneous snapshot of the meteorological conditions over the air quality station instead of a regional average for a given night. We have clarified it in Lines 141-143 and Lines 164-165:

'Since the ERA5 reanalysis dataset was gridded, the nearest-neighbour interpolation method is used to obtain site-specific meteorological variables at the 16 air quality monitoring sites.' (Lines 141-143)

'Low-level jets (LLJs) and convective storms (Conv) are defined in this study based on the above site-specific ERA5 reanalysis dataset.' (Lines 164-165)

(5) We have added Table 1 to summary the dataset used in this study and their purpose as follow:

**Table 1. Summary of the dataset used in this study**

| Product | Period | Sites | Temporal resolution | Spatial resolution | Purpose |
|---|---|---|---|---|---|
| Observed $O_3$ data | 2006-2019 | 16 sites | 1 h | - | Spatiotemporal analysis of NOI and NOP, model performance |
| Observed vertical $O_3$ data | 2019 | Dongguan superstation | 12 min | - | Analysis of an NOI event caused by Conv |
| Observed meteorological data | 2017.09.08-2017.09.15 | 9 sites | 1 h | - | Model performance |
| Observed Cloud-top Temperature (CTT) data | 2019 | Gridded data | 1 h | 0.1° | Indicator of the occurrence of convection |
| ERA5 reanalysis dataset | 2006-2019 | Gridded data | 1 h | 0.25° | Definition of LLJs and Conv |

We have also clarified the purpose of these datasets in the revised manuscript as follows:

'The observed hourly $O_3$ data were used for subsequent NOI and NOP analyses, and evaluation of $O_3$ simulations.' (Lines 122-123)

'In this study, based on the above observed hourly $O_3$ data at the 16 air quality monitoring sites, NOI events are identified at each site, yet only one NOI event is recorded per night, regardless of how many NOI events occur in a single night.' (Lines 158-161)

'The vertical distribution of $O_3$ concentrations observed at the Dongguan superstation (23.02° N, 113.79° E) in 2019 is also used to investigate the impact of Conv on a particular NOI event.' (Lines 124-125)

'The observed meteorological data were used to evaluate the performance of the model.' (Line 131)

'To investigate the impacts of meteorological processes on NOI events, the ERA5 reanalysis dataset (https://cds.climate.copernicus.eu/cdsapp#!/home, last accessed on February 10, 2022) provided by the European Centre for Medium-Range Weather Forecasts (ECMWF) is used in this study.' (Lines 133-135)

'Low-level jets (LLJs) and convective storms (Conv) are defined in this study based on the above site-specific ERA5 reanalysis dataset.' (Lines 164-165)

'The observed cloud-top temperature (CTT) data for 2019 obtained from the Fengyun-2G satellite (http://satellite.nsmc.org.cn/, last accessed on August 31, 2022) are used to indicate the occurrence of convection.' (Lines 144-146)

'Due to the lack of observed vertical profiles of wind speed, the WRF-CMAQ model is employed to investigate the effects of LLJs on a selected NOI event.' (Lines 198-199)

3. In Figure 3 (and all of the accompanying analysis), the authors take data from 2006 – 2019 and break it into two halves, with a breakpoint at 2012 for (a) and (b) and 2016 for (c) and (d). It would be nice to have some physical justification for applying a discontinuity in the linear trend analysis at these specific years. For example, did any local policies on emissions change in 2012 or 2016? If there was no specific justification in mind, the authors should clearly state this section of their research as exploratory and at least speculate on a physical cause, otherwise, this could be seen as "p-hacking". Also, please state the statistical method used for calculating the p-values of the linear trends.

Response:

Thanks for your suggestion.

(1) The breakpoint at 2012 for NOI and LLJs is more likely related to the change of urbanization. We have clarified it in Lines 275-284:

'Both the frequency of NOI and LLJs present increasing trends before 2012 and decreasing trends thereafter, which was related to urbanization. Previous studies have shown that urbanization has large effects on the frequency of LLJs by changing surface conditions (roughness and soil moisture) and further affecting the turbulence and geostrophic wind speed (McCorcle, 1988; Fast and McCorcle, 1990; Kallistratova, 2008; Nikolic et al., 2019; Ziemann et al., 2019). Kallistratova (2008) and Nikolic et al. (2019) pointed out that negative correlation was found between urban areas and the frequency of LLJs. During 1987-2017, the urban areas in the PRD region grew at an average rate of 8.82% yr$^{-1}$ (Yang et al., 2019a) and reached maximum urban land expansion growth rate of 6.66% during 2010-2015 (Zhang et al., 2021). Therefore, the trends for the frequency of NOI and LLJs were quite different during these two periods (2006-2011 and 2012-2019).'

(2) As the reviewer said, the breakpoint at 2016 for NOP and MDA8 $O_3$ was more likely related to the change of precursor emission. We have clarified it in Lines 299-311:

'The variations of NOP and MDA8 $O_3$ during the two periods (2006-2015 and 2016-2019) are more likely related to the change in precursor emissions. The continuous increase in the emissions of anthropogenic VOCs and $NO_x$ resulted in the gradual increase of $O_3$ concentrations between 2006 and 2012 (Ma et al., 2016; Li et al., 2017; Zhong et al., 2018; Liao et al., 2021). However, since the implementation of Air Pollution Prevention and Control Action Plan (APPCAP) in 2013, $NO_x$ emissions was dramatically decreased by 21% in 2017 compared to 2013 (Feng et al., 2019; Yang et al., 2019b). The weakening of NO titration caused by the dramatic decrease in $NO_x$ emissions and the continuously

increasing VOCs emissions due to the lack of controls became important drivers of the sharp rise in $O_3$ since 2015 (Li et al., 2019; Mousavinezhad et al., 2021; Li et al., 2022). Furthermore, the decreasing $PM_{2.5}$ levels and the increasing atmospheric oxidizing capacity in the PRD region in recent years have also been considered as important contributors to accelerated $O_3$ growth during 2016-2019 (Gong et al., 2018; Li et al., 2019; Han et al., 2019). Consequently, NOP and MDA8 $O_3$ present slower increase rate before 2015 and higher increase rate thereafter.'

(3) We used Mann-Kendall test method to calculate the p-value of the linear trends. We have clarified it in Lines 190-196:

'In this study, the nonparametric Mann-Kendall (M-K) test (Mann, 1945) is used to determine the statistical significance (p values) associated with the annual trends of NOI, NOP, MDA8 $O_3$, LLJs and Conv, etc. A significance level of $p < 0.05$ was used to test the significance of the inter-annual trend. The magnitude of a given trend is calculated by the nonparametric Theil-Sen (T-S) estimator (Sen, 1968). The advantage of the M-K test and the T-S estimator is that they do not require prior assumptions of the statistical distribution for the data and are resistant to outliers. The M-K test and the T-S estimator have been widely used in previous $O_3$ trend studies (Wang, et al., 2019; Lu et al., 2020; Li et al., 2022).'

4. I appreciate that the authors recognize the controversy of whether NOI increases or decreases the following days ozone concentration in lines 426-429. In lines 241-243, the authors state that "NOP is significantly positively correlated with MDA8 $O_3$ … implies that daytime $O_3$ concentration levels potentially affect NOP". To further strengthen this discussion on the relationship between daytime and nighttime ozone, I would suggest the authors look at the correlations between: 1) the afternoon MDA8 and the following night's NOP, and 2) the NOP and the following afternoons MDA8, and explicitly report the results from both. This will help elucidate the arrows of causality between daytime and nighttime ozone concentrations in the PRD.

Response:

Thanks for pointing out this critical issue, which is indeed a good point. According to the reviewer's comments, we have analyzed the correlation between the afternoon MDA8 $O_3$ and the following night's NOP (Fig. 12a), and the correlation between NOP and the following afternoon MDA8 $O_3$ (Fig. 12b), respectively. The results show that MDA8 $O_3$ was positively correlated with NOP, suggesting that daytime MDA8 $O_3$ and nighttime NOP affected each other. We have provided more discussion in Lines

'… To further explore the relationship between the daytime MDA8 $O_3$ and nighttime NOP in the PRD region, we display the correlation between the MDA8 $O_3$ and the following night's NOP (shorthand MDA8-NOP) (Fig. 12a) and the NOP and the following MDA8 $O_3$ (shorthand NOP-MDA8) (Fig. 12b), respectively. The results show that MDA8 $O_3$ was positively correlated with NOP with a correlation coefficient of 0.63 (p<0.01) and 0.56 (p<0.01) for MDA8-NOP and NOP-MDA8, respectively, suggesting an interplay between daytime $O_3$ and NOP in the PRD region.'

[Figure]

**Figure 12. Correlation between (a) the afternoon MDA8 $O_3$ concentration and the following night's NOP concentration and (b) the NOP concentration and the following afternoon MDA8 $O_3$ concentration**

5.  As for my most significant concern, the authors use a K index (KI) > 30 as an indication of whether convective storms are occurring in the PRD on a given night. While I understand the need to make approximations when using large datasets, deep convection can occur when KI<30 and KI>30 does not guarantee the presence of deep convection, so there needs to be additional justification that KI>30 is a valid metric for what the authors are trying to capture. For example, the authors might look at a random subset of 10 nights where KI < 30 and 10 nights where KI > 30, and qualitatively compare the radar and/or satellite imagery in the PRD. Alternatively, they could look at the relationship between KI and peak vertical velocity in the ERA5 for the PRD at night, and show that KI=30 is a good cutoff for their purposes.

Response:

Thanks for pointing out this critical issue.

Firstly, apart from KI, cloud-top temperature (CTT) value was further introduced as an indicator of the

occurrence of convective system. A lower CTT value suggest that the probability of convection event is higher. According to the work of Ai et al. (2016), CTT lower than -35 ℃ indicate the occurrence of convection. In addition, we have randomly selected 10 nights with KI > 30 ℃ (Table S3 and Figure S1) and 10 nights with KI < 30 ℃ (Table S4 and Figure S2) and calculated their corresponded CTT values. For the cases with KI > 30 ℃, 10 out of 10 cases were with CTT lower than -35 ℃. And the spatial distribution of CTT shows that the CTT exhibits a distinct circular lower value area over the selected sites, indicating the occurrence of convective system.

For the cases with KI < 30 ℃, 6 out of 10 cases were with CTT higher that -35 ℃, while the rest of 4 cases with no CTT data due to cloudless weather. And the spatial distribution of CTT does not show the features of a convective system, suggesting that convective processes have not been observed for the selected 10 cases with KI < 30 ℃.

The above results suggested that the KI > 30 ℃ criterion is a valid metric to capture the occurrence of convection. We have provided more information in Lines 174-185:

'Cloud-top temperature (CTT) was also introduced as an indicator of the occurrence of convective systems and further used to evaluate the applicability of KI. The lower the CTT, the higher the probability of convection event. According to the work of Ai et al. (2016), CTT lower than -35 ℃ indicates the occurrence of convection. We randomly selected 10 nights with KI > 30 ℃ (Table S3) and 10 nights with KI < 30 ℃ (Table S4) and examined the corresponding CTT values. In the cases with KI > 30 ℃, the CTT values were lower than -35 ℃ in 10 out of 10 nights (Table S3). And the spatial distribution of CTT showed that they had a distinct circular area with lower value over the selected sites, indicating the occurrence of convective systems (Fig. S1). For the cases with KI < 30 ℃, 6 out of 10 nights were with CTT higher than -35 ℃, while the rest 4 nights had no CTT data due to cloudless weather (Table S4). The spatial distribution of CTT did not show the features of a convective system (Fig. S2), suggesting that convection was not observed for the selected 10 cases with KI < 30 ℃. The above results suggest that the KI > 30 ℃ criterion is a valid metric to capture the occurrence of convection.'

**Table S3. Site-specific values of KI and CTT for the randomly selected cases with KI > 30 °C**

| Time (LT) | Site | KI (°C) | CTT (°C) | Figure |
|---|---|---|---|---|
| 2019/04/11 22:00 | WQS | 33 | -49 | Figure S1 (a) |
| 2019/04/16 00:00 | XP | 33 | -45 | Figure S1 (b) |
| 2019/05/26 00:00 | JJJ | 32 | -62 | Figure S1 (c) |
| 2019/06/25 23:00 | DH | 34 | -51 | Figure S1 (d) |
| 2019/07/02 22:00 | NCYL | 35 | -54 | Figure S1 (e) |
| 2019/07/21 21:00 | NCYL | 32 | -47 | Figure S1 (f) |
| 2019/08/08 22:00 | LY | 39 | -70 | Figure S1 (g) |
| 2019/08/24 23:00 | LY | 31 | -68 | Figure S1 (h) |
| 2019/09/14 23:00 | HJC | 31 | -48 | Figure S1 (i) |
| 2019/10/07 00:00 | TJ | 37 | -68 | Figure S1 (j) |

**Table S4. Site-specific values of KI and CTT for the randomly selected cases with KI < 30 °C**

| Time (LT) | Site | KI (°C) | CTT (°C) | Figure |
|---|---|---|---|---|
| 2019/04/12 22:00 | WQS | 29 | 12 | Figure S2 (a) |
| 2019/04/17 00:00 | XP | 11 | Cloudless | Figure S2 (b) |
| 2019/05/03 00:00 | JJJ | 29 | 2 | Figure S2 (c) |
| 2019/06/27 23:00 | DH | 25 | Cloudless | Figure S2 (d) |
| 2019/07/04 22:00 | NCYL | 26 | -3 | Figure S2 (e) |
| 2019/07/25 21:00 | NCYL | 27 | Cloudless | Figure S2 (f) |
| 2019/08/04 22:00 | LY | 26 | -20 | Figure S2 (g) |
| 2019/08/20 23:00 | DH | 29 | -6 | Figure S2 (h) |
| 2019/09/15 23:00 | HJC | 28 | Cloudless | Figure S2 (i) |
| 2019/10/08 00:00 | TJ | 26 | 19 | Figure S2 (j) |

[Figure]

**Figure S1. Spatial distribution of CTT for the randomly selected cases with KI > 30 ℃. (a) to (j) refer to Table S3.**

[Figure]

**Figure S2. Spatial distribution of CTT for the randomly selected cases with KI < 30 °C. (a) to (j) refer to Table S4.**

6. On a related note to (5), the case study of a Conv event presented in section 3.6 could use some additional supporting evidence and data. Figure 9b shows updrafts of only up to 5 cm s⁻¹, which are at least an order of magnitude lower than what would be expected in convective showers and

thunderstorms. While some light precipitation is indicated in Figure 9a, it would be better to also see a radar and/or satellite image of the alleged convective storms that night.

Response:

Thanks for pointing out this critical issue.

(1) We agree with the reviewer that the vertical wind velocity was relatively low, which has also been found in previous studies (Ploeger et al., 2021). Although the vertical wind velocity was relatively low, the vertical velocity results show continuous updraft trends at 1-3 km altitude during the afternoon (Fig. 8b), which still can indicated the happening of convection.

(2) According to the reviewer's comments, we used additional data (cloud-top temperature, CTT) to provide evidence for the occurrence of convection. The spatial distribution of CCT (Fig. 9) showed that the CTT was -66 ℃ over Dongguan, which was lower than the criteria (-35 ℃) for the happening of convection. The CTT results further suggested a high possibility of the happening of convection process.

We have provided the related description in Lines 398-404:

'The KI remains above 36 ℃ (Fig. 8a) and the vertical velocity show continuous updraft trends at 1-3 km altitude from 14:00 to 23:00 (Fig. 8b), indicating a high possibility of convection. Although the magnitude of vertical velocity was relatively low, it also has been found in previous studies (Ploeger et al., 2021). In addition, the spatial distribution of CTT show that the CTT value at 18:00 over Dongguan was around -70 ℃ (Fig. 9a), which was lower than the criterion (-35 ℃) for the happening of convection process. The results of KI, the vertical velocity and the CTT indicate that the possibility of a convection process is high.'

[Figure]

Figure 9. Spatial distribution of cloud-top temperature (CTT) at (a) 18:00 and (b) 21:00 LT on September 3, 2019.

**Specific Comments:**

7. Line 45: Please explicitly introduce the chemical reaction for NO titration for unfamiliar readers.

Response:

Thanks for your suggestion. We have added more descript in Lines 43-46:

'During nighttime, $O_3$ production ceases owing to the absence of sunlight, and dry deposition and NO titration (Eq. (1)) remove $O_3$ directly from the atmosphere, lead to relatively low $O_3$ concentrations at night (Jacob, 2000; Brown et al., 2006).

$$NO + O_3 \rightarrow NO_2 + O_2 \qquad\qquad (1)'$$

8. Line 50: "around 3:00 in the morning" local time or UTC? Please specify.

Response:

It is local time. We have modified it in Line 51:

'… around 3:00 (LT) in the UK …'

9. Line 50: "118 µg m$^{-3}$ in the UK" compared to roughly what values in the daytime?

Response:

Thank you. We compared the value 118µg m$^{-3}$ with monthly average daytime values ($69 \pm 10$ µg m$^{-3}$). We have modified the description in Lines 50-52:

'Kulkarni et al. (2015) found that NOI events were observed around 03:00 (LT) in the UK, with concentrations as high as 118 µg m$^{-3}$, which was much higher than the monthly average daytime $O_3$ concentration ($69 \pm 10$ µg m$^{-3}$).'

10. Lines 51-52: "and the annual trend was found to be increasing" in terms of frequency of occurrence or intensity? Or both?

Response:

The "increasing" refers to the nocturnal $O_3$ concentration. We have modified it in Line 53-54:

'… and the annual trend of nocturnal $O_3$ concentration was found to on the increase …'

11. Lines 52-53: "High nocturnal $O_3$…pollution events" but in lines 426-429 you mention that this is

controversial. Better to be consistent.

Response:

We have deleted the original sentence "High nocturnal $O_3$…pollution events" to avoid confusion, and modified the description in Lines 490-496:

'The occurrence of NOI events is likely to impact the $O_3$ levels on the following day, which makes $O_3$ prevention more complex and challenging (Ravishankara, 2009; Sullivan et al., 2017). However, the relationship between NOI events and the following daytime $O_3$ pollution remains unclear and controversial. Kuang et al. (2011) and Sullivan et al. (2017) revealed that NOI events led to a higher increasing rate of $O_3$ and worse air quality on the following day, while Klein et al. (2019) and Caputi et al. (2019) observed lower $O_3$ levels during the daytime following NOI events.'

12. Line 54: We use the word "proven" in mathematics but not science. "Shown", "suggested", "provided evidence for", or anything similar could be used instead.

Response:

We have replaced "proven" with "shown" throughout the manuscript.

13. Line 63: "With an altitude of about 500 m" this is a general average cited in Stull, how well does this apply to the PRD?

Response:

Sugimoto et al. (2009) had conducted a campaign to observe boundary layer height in the PRD region by using lidar and found that the nocturnal boundary layer height is around 500 m. And Fan et al. (2022) reported a nocturnal boundary height of around 400 m in the PRD region. Therefore, the nocturnal boundary layer height was around 400-500 m in the PRD regions, which was comparable with the average value provided by Stull, (1988). We have modified the description in Lines 62-63:

'… with an altitude of 400-500m (Stull, 1988; Sugimoto et al., 2009; Fan et al., 2022).'

14. Line 69: Please clarify what is meant by "dynamic variation".

Response:

We apologized for our vague description. We have rewritten this sentence in lines 70-72:

'Dias-Junior et al. (2017) revealed that downdrafts induced by Conv play an important role in triggering

NOI events in the Amazon region of Brazil based on 1-yr observations.'

15. Line 71: Please change "Tropospheric" to "Free Tropospheric" because we are distinguishing multiple layers of the troposphere in this study (residual layer, nocturnal boundary layer, free troposphere).

Response:

Modified as suggested.

16. Line 96: "attributable to differences … urbanization". Is this because of the differences in nighttime NO emissions in urban vs. rural areas? Please state.

Response:

We apologized for the vague expression in the original manuscript. We have rewritten this sentence in lines 96-98:

'Tong and Leung (2012) observed a double-peak pattern of diurnal $O_3$ variation in Hong Kong during 1990-2005, and found that nocturnal $O_3$ peaks are sometimes higher than daytime maxima.'

17. Lines 103-104: "Moreover, high daytime … in the PRD region". Please add citation. Also, another motivating factor would probably be the high population in the PRD and the number of people air quality affects in this region?

Response:

We have modified this sentence in line 104-107:

'… In addition, high population densities and increasing number of people active at night in the PRD region make NOI events an important potential risk to human health (Kurt et al., 2016; Carré et al., 2017; Yang et al., 2019a; Zhang et al., 2021).'

18. Figure 1: If it wouldn't create much additional work, it may be worth shading this map with terrain instead of coloring the political regions. The fill colors for the political boundaries don't add much useful information to the plot.

Response:

We have accordingly replotted Figure 1 as follows:

[Figure]

**Figure 1. Model domains and locations of 16 air quality monitoring sites (purple dots), 9 meteorological stations (blue triangles), and Dongguan superstation (red triangles). The underlying figure shows the elevation of the terrain (m).**

19. Line 142: Again, please avoid using the words "proof" or "proven" in a science article.

Response:

We have replaced "proven" with "shown" throughout the manuscript.

20. Line 164: Consider changing "explored" to "utilized"

Response:

We have replaced "explored" with "utilized" in the revised manuscript.

21. Lines 201-205: These statistics are cutoff ranges that the EPA considers a model acceptable or unacceptable to use, correct? Please specify the purpose of introducing these values, statistics, and ranges here – it isn't entire clear.

Response:

The reviewer is correct. These statistics are cutoff ranges that the EPA considers a model acceptable or unacceptable to use. We introduced these statistics to evaluate the model performance by using some specific quantified index. We have modified it in Lines 245-251:

'The evaluation protocols of the U.S. Environmental Protection Agency (EPA, 2017) are used to evaluate the performance of the meteorological parameters. The simulated results were accepted when the

statistics met the criteria listed as follows: MB $\leq \pm 0.5$ °C and IoA $\geq 0.8$ for simulated T2; MB $\leq \pm 5\%$ and IoA $\geq 0.6$ for simulated RH; and MB $\leq \pm 0.5$ m/s, RMSE $\leq 2.0$ m/s, and IoA $\geq 0.6$ for simulated WS10. The evaluation protocols of the Ministry of Environmental Protection of China (MEP, 2015) are used to evaluate the performance of $O_3$ and the simulated results were acceptable if the statistics met the criteria listed below: $-15\% < NMB < 15\%$, $NME < 35\%$, and $r > 0.4$.'

22. Lines 219-220: In order to consider LLJ events a "downdraft", there must be an assumption that the LLJ is inducing turbulent mixing from the vertical wind shear it creates. Please state this.

Response:

We have clarified it in Lines 167-168:

'It's worth noting that the LLJs defined in this study only consider the turbulence mixing induced by their vertical wind shear.'

23. Line 264: "Below" -> please specify "below the jet"

Response:

We have replaced "below" with "below the jet" in Line 323.

'Given that LLJs can enhance turbulence below the jet …'

24. Lines 278 – 279: "it can bring clean marine air into the PRD region" I assume this is at the surface?

Response:

The reviewer is correct. We have modified this sentence in Lines 337-338:

'… This is mainly due to the typical Asian monsoon circulation, which brings clean marine air to the lower troposphere of the PRD region in summer…'

25. Line 297: what is meant exactly by $O_3$ from the daytime "enters" the RL between 21:00 and 03:00? Why isn't the $O_3$ already inside the RL from the minute the daytime boundary layer fades into the RL?

Response:

The reviewer is correct. $O_3$ has already stayed inside the RL instead of entering the RL. We have modified this sentence in Lines 358-359:

'As the sun sets and the daytime boundary layer fades away, the $O_3$ produced during daytime remains at a relatively high level in the RL during 21:00-03:00.'

26. Line 299: Horizontal transport to where? And by "vertical transport" of $O_3$ are the authors referring to dry deposition, convection, or both?

Response:

(1) The horizontal transport means the $O_3$ remained at the RL can be horizontal transported to the downwind area.

(2) The vertical transport includes the vertical exchange caused by LLJs, convection, and the $O_3$ dry deposition processes.

We have modified this sentence in Lines 360-362:

'After 03:00, the $O_3$ concentrations in the RL decreased due to horizontal transport to downwind area and vertical transport (e.g., LLJs, convection, $O_3$ dry deposition process) during 21:00-03:00, …'

27. Line 319: How are individual sites classified as either rural or urban?

Response:

HKEPD (2017) classified the individual sites into urban and rural according to the land use type and surrounding environment of the monitoring stations.

According to the comments #7 provided by the reviewer #2, the spatial distribution of the sites is more important than the type of these sites. Therefore, we only discuss the spatial difference instead of difference between urban and rural, and we have deleted the description associated with the difference between urban and rural in the revised manuscript.

28. Figure 8: Similar to general comment (3), please provide a justification for the break at 2012.

Response:

We have deleted Figure 8 and the associated description in the revised manuscript. Please refer to our detailed responses to the reviewer #2 comments #7.

29. Lines 352-353: the modeled downdraft in Figure 9b occurs *after* the observed $O_3$ intrusion into the nocturnal boundary layer. It may be that the timing of the model is slightly off, but this should

be acknowledged rather than stated as a clear cause an effect.

Response:

Thanks for pointed out this issue. We have acknowledged it in Lines 415-418:

'Although the modeled downdraft occurred at 22:00-23:00 (Fig. 8b) was around half an hour later than the observed $O_3$ intrusion into the nocturnal boundary layer (Fig. 8c) due to the model bias, the modeled results can still generally capture the occurrence of convection processes.'

30. Line 369: "meet the criteria" -> the EPA criteria specified earlier? Please clarify.

Response:

The reviewer is correct. We have clarified it in Lines 431-432:

'The results show that WS10 was reasonably well simulated, as the regional average of MB, RMSE, and IoA met the EPA criteria mentioned in section 2.6.'

31. Figure 10: The black lines (NET) is categorically different from ozone and wind because it is not a meteorological phenomena. This was a bit confusing to my eye at first because it is plotted along with ozone and wind, but in reality it relates more to the bars. Consider at least changing the circle marker to a triangle for the black NET lines.

Response:

According to the reviewers' comments, we have replotted Figure 10 as follows:

[Figure]

**Figure 10. Contribution of individual processes to (a) hourly O₃ concentration near the surface during September 13-14, 2017 and (b) vertical O₃ concentration at 21:00 on September 13, 2017. VTRA: vertical transport, the net effect of vertical advection and diffusion; HTRA: horizontal transport, the net effect of horizontal advection and diffusion; CHEM: gas-phase chemistry; CLDS: cloud processes; DDEP: dry deposition; NET: the net change in O₃ due to all atmospheric processes.**

32. Figure 10b: Would it be possible to plot the momentum flux in the model as well to get an indication of shear below the LLJ? Or no because this was not a large eddy simulation?

Response:

We apologized that we cannot plot the momentum flux since we did not conduct large eddy simulation.

33. Line 428: Caputi et al. 2019 also found lower ozone the following day when more mixing of ozone from the residual layer to nocturnal boundary layer occurred overnight.

Response:

We have modified this sentence in Lines 493-496:

'Kuang et al. (2011) and Sullivan et al. (2017) revealed that NOI events led to a higher increasing rate of O₃ and worse air quality on the following day, while Klein et al. (2019) and Caputi et al. (2019) observed lower O₃ levels during the daytime following NOI events.'

**Technical Comments:**

34. Line 91: "Long-tern" Long-term?

Response:

Modified as suggested.

35. Figure 6b caption: reference to blue but no blue in figure, assume orange?

Response:

Modified as suggested.

**Reference:**

[revised manuscript text omitted]

---

## Author Comment (AC2)

**Responses to Reviewers' comments**

To the esteemed Editor and Reviewers,

We would like to thank the reviewers for the time and efforts in reviewing our manuscript. We have revised the manuscript according to the reviewers' detailed comments, which we sincerely hope the correction will meet with the high publishing standard of the journal. Please find the point-to-point responses to the reviewers' comments as follows:

Reviewers' comments are in black.

Author's responses are in blue color.

Changes in the manuscript are in red color.

Sincerely,

Weihua Chen

On behalf of the authors

**Response to Reviewer #2:**

**Referee #2 comment:**

The study analyzed the spatial and temporal characteristics of nocturnal ozone increase (NOI) events over the Pearl River Delta (PRD) region in southern China. A long-term (2006-2019) record of the NOI events were identified based on surface ozone measurements and meteorology reanalysis, and interpreted with vertical lidar measurements and a regional model simulation. The results showed that low-level jets (LLJs) and convective storms (Conv) were the main drivers of the NOI events. Underlying processes were also analyzed using sample NOI events.

Overall, I think this is a well conducted study, offering quite comprehensive information on the NOI events (trend, seasonal variation, spatial distribution, triggering process) over the PRD region. In particular, the study emphasized the important role of vertical transport, combined with daytime ozone levels for the occurrence of NOI.

The presentation of the manuscript is also clear and in general concise. I recommend publish on ACP after the following comments been addressed.

**Response:**

We thank the reviewer for the positive comments. We have incorporated all your constructive comments and suggestions in the revised manuscript.

**Specific comments:**

**1. Page 5, Section 2.1**

For the definition of NOI, how did you define "remaining for at least 1h" as hourly ozone measurements were used. For example, in Figure 9a, there was a second ozone peak at 0300 am. Would you define this peak as a NOI event?

**Response:**

Thanks for pointing out this critical issue.

The definition of "remaining for at least 1h" means a decrease in  $O_3$  of less than 10 µg m-3 at the next hour.

The second  $O_3$  peak at 03:00 will not be recorded as an NOI event because we record the frequency of NOI events on a unit of day and do not focus on how many NOI events have occurred on a single night. We have clarified the definition of NOI in Lines 155-161: 'For our analysis, we define a nocturnal  $O_3$  increase (NOI) event as  $O_3$  concentrations peaked at night (from 21:00 LT to 06:00 LT the next day), with an increase in levels of at least 10 µg m-3 compared to the previous hour and a decrease of less than 10 µg m-3 in the next hour. The corresponding nighttime peak concentration of  $O_3$  is referred to as the nocturnal  $O_3$  peak (NOP) (Zhu et al., 2020). In this study, based on the above observed hourly  $O_3$  data at the 16 air quality monitoring sites, NOI events are identified at each site, yet only one NOI event is recorded per night, regardless of how many NOI events occur in a single night.'

**2. Page 6, Section 2.2**

The definitions of LLJ and Conv events need some further clarification. In the analysis below, there are LLJs, Conv, and LLJs&Conv events. How did you define LLJs&Conv? Were they also accounted in the LLJs or Conv events?

Response:

Thanks for your suggestion. In our study, NOI events are classified into 4 categories: caused by LLJs only, caused by Conv only, caused by LLJ and Conv at the same time, and caused by other factors. LLJs&Conv in the original manuscript means an NOI event caused by LLJs and Conv at the same time. We have replaced "LLJs&Conv" with "LLJs+Conv" to keep consistency throughout the revised manuscript. We have clarified it in Lines 186-188:

'In this study, an NOI event at each air quality site was classified into four categories: caused by LLJs only, caused by Conv only, caused by LLJs and Conv (LLJs+Conv) at the same time, and caused by other factors.'

3. The title of Section 2.2 "simulated meteorological data"

Suggest change it to "Meteorology reanalysis data", to avoid confusion with the WRF simulated meteorology.

**Response:**

Modified as suggested.

**4. Section 2.4**

The dry deposition process (DDEP) was included in the process analysis, however, when using Figure

10, DDEP was not shown there even for the surface ozone concentration. This seemed to be unclear and unexplained. Can you please check and clarify it?

**Response:**

We apologized for the missing calculation of DDEP in the original manuscript. We have replotted

Figure 10 as follows:

Figure 10. Contribution of individual processes to (a) hourly O3 concentration near the surface during September 13-14, 2017 and (b) vertical O3 concentration at 21:00 on September 13, 2017. VTRA: vertical transport, the net effect of vertical advection and diffusion; HTRA: horizontal transport, the net effect of horizontal advection and diffusion; CHEM: gas-phase chemistry; CLDS: cloud processes; DDEP: dry deposition; NET: the net change in O3 due to all atmospheric processes.

5. Line 438-440, and in Abstract.

How about the 16% of NOI events attributed to LLJs&Conv? Were these events triggered by LLJs or Conv?

**Response:**

16% of NOI events were caused by LLJs and Conv at the same time. We have modified the description in Lines 19-20 and Lines 511-512:

'Low-level jets (LLJs) are the main meteorological processes triggering NOI events, explaining on average 61% of NOI events.' (Lines 19-20)

'LLJs are the dominant factors causing NOI events (61%), followed by the combination of LLJs and Conv (LLJs+Conv) with a value of 16%.' (Lines 511-512)

6. Page 14, Section 3.5

In Figure 7, there were three stations in the southern part that showed very high NOI frequencies, while the LLJs frequencies were low there. Can you explain and discuss this feature?

Response:

Thanks for pointing out this important issue. Highest frequency of NOI events for the three sites located in the southern part of the PRD regions were also affected by non-LLJs (=Conv+(LLJs+Conv)+Other) because the contributions of LLJs and non-LLJs to the NOI events were comparable at these three sites, while the contributions of LLJs (60-70%) were higher than that of non-LLJs at the rest of sites (Table S5). We have provided more descriptions in Lines 383-388:

'At the three sites located in the southern part of the PRD regions (TJ, TW, and TC), the frequency of NOI events was the highest while the frequency of LLJs was not. This is because these three sites were also affected by non-LLJs (=Conv+(LLJs+Conv)+Other) processes with comparable contributions of LLJs and non-LLJs to the NOI events. And the contributions of LLJs (60-70%) were higher than those of non-LLJs at the rest of sites (Table S5).'

7. Section 3.5, Figure 8

It is not clear that why analyzing the difference between urban and rural areas could elaborate the effect of meteorological process. Based on their locations (Figure 7a), the difference between urban and rural areas may not reflect their urban vs. rural land cover, but their different spatial locations. Please clarify. Response:

We agree with the reviewer that analyzing the difference between urban and rural areas cannot elaborate the effect of meteorological. We have deleted this paragraph and maintained spatial analysis in the revised manuscript.

**Technical comments:**

8. Line 300, "Hance" should be "Hence" Response:

Modified as suggested.

9. Line 308, figure caption, "blue" should be "orange"

Response:

Modified as suggested.

10. Line 425, "improve the next day chemical budget". The word "improve" is misleading here.

Response:

We have replaced "improve" with "impact" in the revised manuscript.

**Reference:**

Zhu, X. W., Ma, Z. Q., Li, Z. M., Wu, J., Guo, H., Yin, X. M., Ma, X. H., and Qiao, L.: Impacts of meteorological conditions on nocturnal surface ozone enhancement during the summertime in Beijing, Atmos. Environ., 225, 117368, https://doi.org/10.1016/j.atmosenv.2020.117368, 2020.

---

## Author Response (AR1)

**Responses to Reviewers' comments**

To the esteemed Editor and Reviewers,

We would like to thank the reviewers for the time and efforts in reviewing our manuscript. We have revised the manuscript according to the reviewers' detailed comments, which we sincerely hope the correction will meet with the high publishing standard of the journal. Please find the point-to-point responses to the reviewers' comments as follows:

Reviewers' comments are in black.

Author's responses are in blue color.

Changes in the manuscript are in red color.

Sincerely,

Weihua Chen

On behalf of the authors

**Response to Reviewer #1:**

**Referee #1 comment:**

Wu et al. provides a novel framework for capturing the essence of nocturnal ozone increases (NOI), which is an important area of research often neglected in analysis of the ozone budget. They break the causes of NOI down into clearly discernable phenomena and present evidence that the majority of NOI events in the Pearl River Delta (PRD) are caused by Low Level Jets (LLJ). Further, they present a detailed case study of a LLJ NOI event, as well as a convective storm (Conv) NOI event.

The framework is intriguing and presents a valuable contribution to the literature, however, some modifications should be made prior to publication. In particular, the authors need to more clearly define their methodology and make a stronger case for using the K index as a proxy for convective storm events. Response:

We thank the reviewer for the positive comments. We have revised our paper according to your comments as follow:

We have reorganized the **Data and methods** section and provided clearer description regarding methodology used in this study. Please refer to our detailed responses to comments #1, #2, #3, and #5.
 We have introduced an additional indicator (cloud-top temperature, CTT) to prove the applicability of K index. Please refer to our detailed responses to comments #5.

**General Comments:**

 The definition of NOI first appears in lines 118-120, where values increase by at least 10 μg m-3. However, this comes from a 2020 reference and not every preceding study in the introduction that mentions NOI contains findings that are consistent with this strict definition (e.g. Caputi et al. 2019). It would be helpful if the authors clarify that (I assume) this is the definition they employ specifically in their study (e.g. "for our analysis, we define NOI as …"). Additionally, the definition needs to be clearer. For example, I am left unsure whether "values increasing by at least 10 μg m-3" mean increasing from the daytime minimum, the previous hour's value, or what exactly.

**Response:**

According to the reviewer's comments, we have provided a specific definition of NOI used in this study in Lines 155-158:

'For our analysis, we define a nocturnal O3 increase (NOI) event as O3 concentrations peaked at night

(from 21:00 LT to 06:00 LT the next day), with an increase in levels of at least 10  $\mu$ g m-3 compared to the previous hour and a decrease of less than 10  $\mu$ g m-3 in the next hour. The corresponding nighttime peak concentration of O3 is referred to as the nocturnal O3 peak (NOP) (Zhu et al., 2020).'

2. Some aspects of the methodology need clarification. In Section 3.1, are the statistics (e.g. 53 +/- 16 d yr-1) from an aggregate of the air quality monitoring stations, and are the error values and error bars in Figures 3 and 4 calculated by a pooled standard deviation? The authors then discuss the proportions of events attributable to LLJ vs. Conv, does this come from the ERA5 data? If so, are the causes of an individual NOI event (LLJ, Conv, LLJ+Conv, Other) determined by an instantaneous snapshot of the meteorological conditions over the air quality station, or a regional average for a given night? Please connect the dots between the different methods discussed (e.g. CMAQ, IPR, air quality stations, meteorological stations, ERA5) and where specifically they employed in the results.

**Response:**

(1) The statistic of average annual frequency of NOI events  $(53 \pm 16 \text{ d yr}^{-1})$  is an average value of the 16 air quality monitoring stations. We have clarified it in Lines 161-162:

'In addition, the regional values of NOI and NOP from the 16 air quality monitoring sites were averaged.'

- (2) The error values and error bars in Figures 3 and 4 indicate the range of deviations for the different station. We have clarified it in Line 317, Line 344 and Line 373:
- '... The error bars indicate the range of deviations for the 16 air quality sites.'
- (3) The proportions of NOI events attributable to LLJs vs. Conv are calculated based on the ERA5 data.We have clarified it in Lines 164-165:

'Low-level jets (LLJs) and convective storms (Conv) are defined in this study based on the above sitespecific ERA5 reanalysis dataset.'

(4) The causes of an individual NOI event are determined by an instantaneous snapshot of the meteorological conditions over the air quality station instead of a regional average for a given night. We have clarified it in Lines 141-143 and Lines 164-165:

'Since the ERA5 reanalysis dataset was gridded, the nearest-neighbour interpolation method is used to obtain site-specific meteorological variables at the 16 air quality monitoring sites.' (Lines 141-143) 'Low-level jets (LLJs) and convective storms (Conv) are defined in this study based on the above site-specific ERA5 reanalysis dataset.' (Lines 164-165)

| (  | 5) | We have added Table | 1 to summar | v the dataset u | sed in this study | and their pu | rpose as follow: |
|----|----|---------------------|-------------|-----------------|-------------------|--------------|------------------|
| t. | 21 |                     | 1 to Summar | y me dataset a  | sou in uns study  | and men pu   | ipose as ionow.  |

| Product                                      | Period                    | Sites                 | Temporal resolution | Spatial resolution | Purpose                                                   |
|----------------------------------------------|---------------------------|-----------------------|---------------------|--------------------|-----------------------------------------------------------|
| Observed O3 data                             | 2006-2019                 | 16 sites              | 1 h                 | -                  | Spatiotemporal analysis of NOI and NOP, model performance |
| Observed vertical O 3 data        | 2019                      | Dongguan superstation | 12 min              | -                  | Analysis of an NOI event
caused by Conv                |
| Observed meteorological data                 | 2017.09.08-
2017.09.15 | 9 sites               | 1 h                 | -                  | Model performance                                         |
| Observed Cloud-top
Temperature (CTT) data | 2019                      | Gridded data          | 1 h                 | 0.1°               | Indicator of the occurrence of convection                 |
| ERA5 reanalysis dataset                      | 2006-2019                 | Gridded data          | 1 h                 | 0.25°              | Definition of LLJs and Conv                               |

**Table 1. Summary of the dataset used in this study**

We have also clarified the purpose of these datasets in the revised manuscript as follows:

'The observed hourly O3 data were used for subsequent NOI and NOP analyses, and evaluation of O3 simulations.' (Lines 122-123)

'In this study, based on the above observed hourly  $O_3$  data at the 16 air quality monitoring sites, NOI events are identified at each site, yet only one NOI event is recorded per night, regardless of how many NOI events occur in a single night.' (Lines 158-161)

'The vertical distribution of O3 concentrations observed at the Dongguan superstation (23.02° N, 113.79°

E) in 2019 is also used to investigate the impact of Conv on a particular NOI event.' (Lines 124-125)

'The observed meteorological data were used to evaluate the performance of the model.' (Line 131)

'To investigate the impacts of meteorological processes on NOI events, the ERA5 reanalysis dataset (https://cds.climate.copernicus.eu/cdsapp#!/home, last accessed on February 10, 2022) provided by the European Centre for Medium-Range Weather Forecasts (ECMWF) is used in this study.' (Lines 133-135) 'Low-level jets (LLJs) and convective storms (Conv) are defined in this study based on the above site-specific ERA5 reanalysis dataset.' (Lines 164-165)

'The observed cloud-top temperature (CTT) data for 2019 obtained from the Fengyun-2G satellite (http://satellite.nsmc.org.cn/, last accessed on August 31, 2022) are used to indicate the occurrence of convection.' (Lines 144-146)

'Due to the lack of observed vertical profiles of wind speed, the WRF-CMAQ model is employed to investigate the effects of LLJs on a selected NOI event.' (Lines 198-199)

3. In Figure 3 (and all of the accompanying analysis), the authors take data from 2006 – 2019 and break it into two halves, with a breakpoint at 2012 for (a) and (b) and 2016 for (c) and (d). It would be nice to have some physical justification for applying a discontinuity in the linear trend analysis at these specific years. For example, did any local policies on emissions change in 2012 or 2016? If there was no specific justification in mind, the authors should clearly state this section of their research as exploratory and at least speculate on a physical cause, otherwise, this could be seen as "p-hacking". Also, please state the statistical method used for calculating the p-values of the linear trends.

**Response:**

**Thanks for your suggestion.**

 The breakpoint at 2012 for NOI and LLJs is more likely related to the change of urbanization. We have clarified it in Lines 275-284:

'Both the frequency of NOI and LLJs present increasing trends before 2012 and decreasing trends thereafter, which was related to urbanization. Previous studies have shown that urbanization has large effects on the frequency of LLJs by changing surface conditions (roughness and soil moisture) and further affecting the turbulence and geostrophic wind speed (McCorcle, 1988; Fast and McCorcle, 1990; Kallistratova, 2008; Nikolic et al., 2019; Ziemann et al., 2019). Kallistratova (2008) and Nikolic et al. (2019) pointed out that negative correlation was found between urban areas and the frequency of LLJs. During 1987-2017, the urban areas in the PRD region grew at an average rate of 8.82% yr-1 (Yang et al., 2019a) and reached maximum urban land expansion growth rate of 6.66% during 2010-2015 (Zhang et al., 2021). Therefore, the trends for the frequency of NOI and LLJs were quite different during these two periods (2006-2011 and 2012-2019).'

(2) As the reviewer said, the breakpoint at 2016 for NOP and MDA8 O3 was more likely related to the change of precursor emission. We have clarified it in Lines 299-311:

'The variations of NOP and MDA8  $O_3$  during the two periods (2006-2015 and 2016-2019) are more likely related to the change in precursor emissions. The continuous increase in the emissions of anthropogenic VOCs and NOx resulted in the gradual increase of O3 concentrations between 2006 and 2012 (Ma et al., 2016; Li et al., 2017; Zhong et al., 2018; Liao et al., 2021). However, since the implementation of Air Pollution Prevention and Control Action Plan (APPCAP) in 2013, NOx emissions was dramatically decreased by 21% in 2017 compared to 2013 (Feng et al., 2019; Yang et al., 2019b). The weakening of NO titration caused by the dramatic decrease in NOx emissions and the continuously increasing VOCs emissions due to the lack of controls became important drivers of the sharp rise in O3 since 2015 (Li et al., 2019; Mousavinezhad et al., 2021; Li et al., 2022). Furthermore, the decreasing PM2.5 levels and the increasing atmospheric oxidizing capacity in the PRD region in recent years have also been considered as important contributors to accelerated O3 growth during 2016-2019 (Gong et al., 2018; Li et al., 2019; Han et al., 2019). Consequently, NOP and MDA8 O3 present slower increase rate before 2015 and higher increase rate thereafter.'

(3) We used Mann-Kendall test method to calculate the p-value of the linear trends. We have clarified it in Lines 190-196:

'In this study, the nonparametric Mann-Kendall (M-K) test (Mann, 1945) is used to determine the statistical significance (p values) associated with the annual trends of NOI, NOP, MDA8 O3, LLJs and Conv, etc. A significance level of p < 0.05 was used to test the significance of the inter-annual trend. The magnitude of a given trend is calculated by the nonparametric Theil-Sen (T-S) estimator (Sen, 1968). The advantage of the M-K test and the T-S estimator is that they do not require prior assumptions of the statistical distribution for the data and are resistant to outliers. The M-K test and the T-S estimator have been widely used in previous O3 trend studies (Wang, et al., 2019; Lu et al., 2020; Li et al., 2022).'

4. I appreciate that the authors recognize the controversy of whether NOI increases or decreases the following days ozone concentration in lines 426-429. In lines 241-243, the authors state that "NOP is significantly positively correlated with MDA8 O3 ... implies that daytime O3 concentration levels potentially affect NOP". To further strengthen this discussion on the relationship between daytime and nighttime ozone, I would suggest the authors look at the correlations between: 1) the afternoon MDA8 and the following night's NOP, and 2) the NOP and the following afternoons MDA8, and explicitly report the results from both. This will help elucidate the arrows of causality between daytime and nighttime ozone concentrations in the PRD.

**Response:**

Thanks for pointing out this critical issue, which is indeed a good point. According to the reviewer's comments, we have analyzed the correlation between the afternoon MDA8 O3 and the following night's NOP (Fig. 12a), and the correlation between NOP and the following afternoon MDA8 O3 (Fig. 12b), respectively. The results show that MDA8 O3 was positively correlated with NOP, suggesting that daytime MDA8 O3 and nighttime NOP affected each other. We have provided more discussion in Lines

496 - 501:

"... To further explore the relationship between the daytime MDA8  $O_3$  and nighttime NOP in the PRD region, we display the correlation between the MDA8  $O_3$  and the following night's NOP (shorthand MDA8-NOP) (Fig. 12a) and the NOP and the following MDA8  $O_3$  (shorthand NOP-MDA8) (Fig. 12b), respectively. The results show that MDA8  $O_3$  was positively correlated with NOP with a correlation coefficient of 0.63 (p

Figure 12. Correlation between (a) the afternoon MDA8 O3 concentration and the following night's NOP concentration and (b) the NOP concentration and the following afternoon MDA8 O3 concentration

5. As for my most significant concern, the authors use a K index (KI) > 30 as an indication of whether convective storms are occurring in the PRD on a given night. While I understand the need to make approximations when using large datasets, deep convection can occur when KI<30 and KI>30 does not guarantee the presence of deep convection, so there needs to be additional justification that KI>30 is a valid metric for what the authors are trying to capture. For example, the authors might look at a random subset of 10 nights where KI < 30 and 10 nights where KI > 30, and qualitatively compare the radar and/or satellite imagery in the PRD. Alternatively, they could look at the relationship between KI and peak vertical velocity in the ERA5 for the PRD at night, and show that KI=30 is a good cutoff for their purposes.

**Response:**

Thanks for pointing out this critical issue.

Firstly, apart from KI, cloud-top temperature (CTT) value was further introduced as an indicator of the

occurrence of convective system. A lower CTT value suggest that the probability of convection event is higher. According to the work of Ai et al. (2016), CTT lower than -35 °C indicate the occurrence of convection. In addition, we have randomly selected 10 nights with KI > 30 °C (Table S3 and Figure S1) and 10 nights with KI < 30 °C (Table S4 and Figure S2) and calculated their corresponded CTT values. For the cases with KI > 30 °C, 10 out of 10 cases were with CTT lower than -35 °C. And the spatial distribution of CTT shows that the CTT exhibits a distinct circular lower value area over the selected sites, indicating the occurrence of convective system.

For the cases with KI < 30 °C, 6 out of 10 cases were with CTT higher that -35 °C, while the rest of 4 cases with no CTT data due to cloudless weather. And the spatial distribution of CTT does not show the features of a convective system, suggesting that convective processes have not been observed for the selected 10 cases with KI < 30 °C.

The above results suggested that the KI > 30 °C criterion is a valid metric to capture the occurrence of convection. We have provided more information in Lines 174-185:

'Cloud-top temperature (CTT) was also introduced as an indicator of the occurrence of convective systems and further used to evaluate the applicability of KI. The lower the CTT, the higher the probability of convection event. According to the work of Ai et al. (2016), CTT lower than -35 °C indicates the occurrence of convection. We randomly selected 10 nights with KI > 30 °C (Table S3) and 10 nights with KI < 30 °C (Table S4) and examined the corresponding CTT values. In the cases with KI > 30 °C, the CTT values were lower than -35 °C in 10 out of 10 nights (Table S3). And the spatial distribution of CTT showed that they had a distinct circular area with lower value over the selected sites, indicating the occurrence of convective systems (Fig. S1). For the cases with KI < 30 °C, 6 out of 10 nights were with CTT higher than -35 °C, while the rest 4 nights had no CTT data due to cloudless weather (Table S4). The spatial distribution of CTT did not show the features of a convective system (Fig. S2), suggesting that convection was not observed for the selected 10 cases with KI < 30 °C. The above results suggest that the KI > 30 °C criterion is a valid metric to capture the occurrence of convection.'

Table S3. Site-specific values of KI and CTT for the randomly selected cases with KI > 30 °C

| Time (LT)        | Site | KI (°C) | CTT (°C) | Figure        |
|------------------|------|---------|----------|---------------|
| 2019/04/11 22:00 | WQS  | 33      | -49      | Figure S1 (a) |
| 2019/04/16 00:00 | XP   | 33      | -45      | Figure S1 (b) |
| 2019/05/26 00:00 | JJJ  | 32      | -62      | Figure S1 (c) |

| 2019/06/25 23:00 | DH   | 34 | -51 | Figure S1 (d) |  |
|------------------|------|----|-----|---------------|--|
| 2019/07/02 22:00 | NCYL | 35 | -54 | Figure S1 (e) |  |
| 2019/07/21 21:00 | NCYL | 32 | -47 | Figure S1 (f) |  |
| 2019/08/08 22:00 | LY   | 39 | -70 | Figure S1 (g) |  |
| 2019/08/24 23:00 | LY   | 31 | -68 | Figure S1 (h) |  |
| 2019/09/14 23:00 | HJC  | 31 | -48 | Figure S1 (i) |  |
| 2019/10/07 00:00 | TJ   | 37 | -68 | Figure S1 (j) |  |
|                  |      |    |     |               |  |

Table S4. Site-specific values of KI and CTT for the randomly selected cases with KI

---

## Referee Report (RR1)

The quality of the manuscript has significantly improved, and I am overall pleased with the authors responses to my comments. I would be happy to recommend the article for publication after the following outstanding issues are addressed.

**General Comments:**

For my general comments, I will retain the same numbering as in my first review.

1. I am still left a little bit confused as to the exact definition of a NOI event. Let's suppose the following are hourly time-series data of ozone concentration in μg m$^{-3}$ within the 21:00 – 6:00 LT range. Please verify that my understanding is correct:

43 54 51 40 = NOI, the concentration increased by >10 (43 to 54) and subsequently decreased by < 10 (54 to 51)

43 54 65 40 = NOI, the concentration increased by >10 (43 to 54) and subsequently didn't decrease (54 to 65) even though in the following hour it decreased by > 10 (65 to 40)

43 47 54 51 = Not NOI, the increase of > 10 happens over more than 1 hour

43 54 42 41 = Not NOI, even though there's an increase of > 10 (43 to 54) there is a subsequent decrease of > 10 (54 to 42)

Also, I assume that the "previous hour" and "next hour" can include 20:00 and 7:00 LT, respectively?

2. The methodology is much clearer in this version, and I have no additional concerns.

3. I appreciate the authors relating the breakpoints of 2012 and 2016 to changes in urbanization and policy. Given the ambiguity of the exact years that these changes went into effect, I do think that one could still make a case that choosing the specific years of 2012 and 2016 to split the data is a form of p-hacking, i.e., finding breakpoints that would make the trends statistically significant rather than strictly testing a pre-determined hypothesis. However, exploratory analyses are an important component of observational field studies and distinguish this type of research from controlled laboratory experiments, where statistical controls are much more rigorous but creativity is highly limited and the scope of work is more narrow. I believe the trends found in this study are important findings to report.

My only additional recommendation on this point is to change "was related to urbanization" to "was likely related to urbanization".

4. The authors did excellent work in following my suggestion to plot MDA8-NOP and NOP-MDA8, and the results are highly intriguing. I agree with their conclusion that it suggests a complex interplay between daytime and nighttime dynamics.

5. Based on the new analyses presented, I accept the use of cloud top temperature as a proxy for deep convection, and the authors now make a satisfactory case that KI is a valid metric of Conv events in the PRD.

6. I appreciate the Conv case study being supplemented with CTT. I also appreciate the citation of Ploeger et al. (2021) showing comparable vertical velocity. It may be that the WRF model cannot resolve the core updrafts in deep convection with 3 km resolution, or that the convective scheme used may not

be designed to do this – though I am not a modelling expert. Regardless, the overall picture of convection occurring on this night is clear with the WRF, CTT, and KI taken as a whole.

**Specific Comments:**

Lines 167-168 (originally #22 - lines 219-220):

I apologize that my original comment was likely unclear. I was suggesting the authors clarify why LLJs can be considered downdrafts for unfamiliar readers. A sentence such as "We assume that LLJs cause downdrafts because of the vertical wind shear the jets induce, which creates mechanical turbulence" would suffice.

Lines 415-418 (originally #29 - lines 352-353):

I accept the authors correction, except I am not sure that "model bias" is the correct term. Would "model uncertainty" or "model imperfection" be more appropriate?

---

## Author Response (AR2)

**Responses to Reviewers' comments**

To the esteemed Editor Dr. Jerome Brioude and Reviewer Dr. Dani Caputi,

We would like to thank Dr. Jerome Brioude for handing this paper. We would also like to thank Dr. Dani Caputi for the time and efforts in reviewing our manuscript. We have accordingly revised the manuscript.

Please find the point-to-point responses to the reviewer' comments as follows:

Reviewer' comments are in black.

Author's responses are in blue color.

Changes in the manuscript are in red color.

Sincerely,

Weihua Chen

On behalf of the authors

**Response to Reviewer #1:**

**Referee #1 comment:**

The quality of the manuscript has significantly improved, and I am overall pleased with the authors responses to my comments. I would be happy to recommend the article for publication after the following outstanding issues are addressed.

Response:

We thank the reviewer's positive comments. We have incorporated all your constructive comments and suggestions in the revised manuscript.

**General Comments:**

For my general comments, I will retain the same numbering as in my first review.

1. I am still left a little bit confused as to the exact definition of a NOI event. Let's suppose the following are hourly time-series data of ozone concentration in $\mu g\ m^{-3}$ within the 21:00 – 6:00 LT range. Please verify that my understanding is correct:

43 54 51 40 = NOI, the concentration increased by >10 (43 to 54) and subsequently decreased by < 10 (54 to 51)

43 54 65 40 = NOI, the concentration increased by >10 (43 to 54) and subsequently didn't decrease (54 to 65) even though in the following hour it decreased by > 10 (65 to 40)

43 47 54 51 = Not NOI, the increase of > 10 happens over more than 1 hour

43 54 42 41 = Not NOI, even though there's an increase of > 10 (43 to 54) there is a subsequent decrease of > 10 (54 to 42)

Also, I assume that the "previous hour" and "next hour" can include 20:00 and 7:00 LT, respectively?

Response:

The reviewer is correct. As the reviewer said, the "previous hour" and "next hour" include 20:00 and 7:00 LT, respectively. We have clarified it in Lines 157-158:

'…, with an increase in levels of at least 10 $\mu g\ m^{-3}$ compared to the previous hour (include 20:00 LT) and a decrease of less than 10 $\mu g\ m^{-3}$ in the next hour (include 07:00 LT).'

2. The methodology is much clearer in this version, and I have no additional concerns.

Response:

Many thanks.

3. I appreciate the authors relating the breakpoints of 2012 and 2016 to changes in urbanization and policy. Given the ambiguity of the exact years that these changes went into effect, I do think that one could still make a case that choosing the specific years of 2012 and 2016 to split the data is a form of p-hacking, i.e., finding breakpoints that would make the trends statistically significant rather than strictly testing a pre-determined hypothesis. However, exploratory analyses are an important component of observational field studies and distinguish this type of research from controlled laboratory experiments, where statistical controls are much more rigorous but creativity is highly limited and the scope of work is more narrow. I believe the trends found in this study are important findings to report.

My only additional recommendation on this point is to change "was related to urbanization" to "was likely related to urbanization".

Response:

Thanks for bringing this issue to our attention. We have modified it in Line 276:

'…, which was likely related to urbanization.'

4. The authors did excellent work in following my suggestion to plot MDA8-NOP and NOP-MDA8, and the results are highly intriguing. I agree with their conclusion that it suggests a complex interplay between daytime and nighttime dynamics.

Response:

Thanks for providing this constructive suggestion to make the results clearer and more convincing.

5. Based on the new analyses presented, I accept the use of cloud top temperature as a proxy for deep convection, and the authors now make a satisfactory case that KI is a valid metric of Conv events in the PRD.

Response:

Many thanks.

6. I appreciate the Conv case study being supplemented with CTT. I also appreciate the citation of Ploeger et al. (2021) showing comparable vertical velocity. It may be that the WRF model cannot resolve the core

updrafts in deep convection with 3 km resolution, or that the convective scheme used may not be designed to do this – though I am not a modelling expert. Regardless, the overall picture of convection occurring on this night is clear with the WRF, CTT, and KI taken as a whole.

Response:

Thanks for the reviewer's approval.

**Specific Comments:**

Lines 167-168 (originally #22 - lines 219-220):

I apologize that my original comment was likely unclear. I was suggesting the authors clarify why LLJs can be considered downdrafts for unfamiliar readers. A sentence such as "We assume that LLJs cause downdrafts because of the vertical wind shear the jets induce, which creates mechanical turbulence" would suffice.

Response:

According to the reviewer's suggestion, we have modified this sentence in lines 167-168:

'We assume that LLJs cause downdrafts because of the vertical wind shear the jets induce, which creates mechanical turbulence'

Lines 415-418 (originally #29 - lines 352-353):

I accept the authors correction, except I am not sure that "model bias" is the correct term. Would "model uncertainty" or "model imperfection" be more appropriate?

Response:

Thanks for pointing out this issue. We intended to say that the modeled downdraft was half an hour later than the observed $O_3$ intrusion, which was related to the model errors. Therefore, "model errors" would be more appropriate and we have modified it in Line 417:

'… due to the model errors, …'